# MEMORIZATION CAPACITY OF MULTI-HEAD ATTENTION IN TRANSFORMERS

**Sadegh Mahdavi[1,2], Renjie Liao[1,2], Christos Thrampoulidis[1]**
[1]University of British Columbia
[2]Vector Institute for AI
`{smahdavi,rjliao,cthrampo}@ece.ubc.ca`

## ABSTRACT

Transformers have become the go-to architecture for language and vision tasks, yet their theoretical properties, especially memorization capacity, remain elusive. This paper investigates the memorization abilities of multi-head attention mechanisms, examining how many example sequences they can memorize, as a function of the number of heads and sequence length. Motivated by experimental findings on vision transformers, we introduce novel assumptions about the linear independence of input data, distinct from the commonly used general-position assumption. Under these assumptions, we demonstrate that an attention layer with $H$ heads, dimension $d$, and context size $n < d$, featuring $\Theta(Hd^2)$ parameters, can memorize $\Omega(Hn)$ examples. Our analysis sheds light on how different attention heads handle various example sequences, aided by the softmax operator's saturation property. We validate our findings through experiments on synthetic data.

## 1 INTRODUCTION

Transformers have emerged as the de-facto architecture for achieving state-of-the-art performance across different domains such as natural language processing and vision tasks, as demonstrated by recent breakthroughs, *e.g.* (Dehghani et al., 2023; Touvron et al., 2023; OpenAI, 2023). These models are characterized by their immense scales, often comprising billions of parameters to achieve their high performances. Given their sizes and complexities, a natural question arises: How effectively can they memorize the training data? This query is important both from a privacy perspective (Carlini et al., 2020), and as a stepping stone towards quantifying the model's ability to generalize to new data (Zhang et al., 2017). Additionally, studying how the memorization of transformers emerges can offer insights into the distinct roles played by their different building blocks.

Concretely, the memorization capacity of a model $f_{\mathcal{W}}$ refers to the minimal set of parameters $\mathcal{W}$ such that $f_{\mathcal{W}}(\boldsymbol{x}) = \boldsymbol{y}$ holds for all input-output pairs $(\boldsymbol{x}, \boldsymbol{y})$ in the training dataset.[1] Historically, this question has been extensively analyzed for fully connected networks (FCNs), dating back to at least the 1980s when Baum (1988) proved the first set of results for *single-hidden-layer* Threshold networks with binary labels. The best known bound on the memorization capacity of single-hidden-layer ReLU FCNs has only recently been established by Bubeck et al. (2020), following numerous recent studies, *e.g.* (Zhang et al., 2017; Yun et al., 2019). Recent studies have also extensively considered extensions to deeper models, *e.g.* (Vershynin, 2020; Park et al., 2021; Vardi et al., 2022; Yun et al., 2019). However, as of now, the best known results consist of order-wise bounds, involving unknown constants and logarithmic factors. The majority of these works on memorization of FCNs are established for continuous inputs assumed in "General Position," a relatively mild assumption, as it holds true under scenarios like small random perturbations of input examples.

In contrast to FCNs, understanding the memorization capacity of transformers is still in its infancy. A common Transformer architecture consists of two main components: (i) a Multi-head Attention (MHA) module and (ii) a ReLU fully-connected network. The MHA module creates convex combinations of its input elements by computing softmax similarities between input representations. This

---

[1]The definition of memorization in this context is distinct from concepts such as "associative memory" Demircigil et al. (2017); Ramsauer et al. (2020) and "neural memory" Sukhbaatar et al. (2015); Geva et al. (2020) which involve associating an input pattern with a given query and have been recently studied for transformers.

operation, unique to transformers when compared to FCNs, is leveraged in practice for its ability to selectively attend to information from various representations and effectively process long sequences (Vaswani et al., 2017). Yet, its theoretical underpinnings remain largely unexplored. This paper aims to contribute towards closing this gap by investigating the memorization capacity of the MHA module. To isolate its role from the ReLU FCN, we take a step toward analyzing Attention-only models.

Concretely, we study a single-layer MHA module with $H$ heads that takes as inputs a $n \times d$ key/context matrix comprising representations of $n$ tokens, and a $d \times 1$ query vector. We are particularly interested in understanding how the number of attention heads, and the context size of the input affect the memorization in this architecture. Our contributions are as follows:

• We introduce a new set of input-data assumptions that are more relaxed compared to General Position assumptions. Our first assumption requires the set of all query vectors to have Kruskal rank at least $n$, while our second assumption needs context vectors of each example to be linearly independent. These are motivated and validated by experiments on real-world models, such as ViT, following a single layer, whether it be Attention-only or a complete Transformer, that is either randomly initialized or pretrained. In contrast, General-Position does *not* hold in either of these settings.

• For this set of practically relevant assumptions, we prove that a single-layer MHA module with $H$ heads, embedding dimension $d$, key/query dimension $d_h$, value dimension $d_v$, context size $n < d$, output dimension $d_{\text{out}} \le d_v$, equipped with a total of $O(Hd(d_h + d_v))$ trainable parameters, is capable of memorizing $\Omega\left(H \min(n, d_h)\right)$ input examples. Specifically when $d_{\text{out}} = d_h = d$ and $n = \Theta(d)$, this finding recovers the optimal order of real-valued numbers that can be memorized with the given number of parameters.

• Our proof illuminates how different attention heads play distinct roles in memorizing different sets of examples, and how this role allocation is facilitated by the softmax. To show memory capacity increases linearly with $H$, our idea is to make each head individually "responsible" for memorizing $\min(n, d_h) - 1$ examples. By leveraging the saturation property of softmax and introducing specific key-query weight adjustments, we design desired softmax logits for individual heads while ensuring minimal interference with other examples.

• In addition to experiments validating our assumptions on the input sequences, we design and discuss synthetic experiments corroborating our findings: namely, the impact of head number $H$, context length $n$ and key/query dimension $d_h$ on memorization, as well as, the occurrence of the saturation property of the softmax operator.

## 2 RELATED WORK

**Theoretical understanding of Transformers.** Driven by the success of Transformers, there have been several attempts at understanding their underlying mechanisms, *e.g.*, (Li et al., 2023b; Luo et al., 2022; von Oswald et al., 2022; Dong et al., 2021; Edelman et al., 2022; Jelassi et al., 2022; Li et al., 2023a; Sahiner et al., 2022). Closer to us, (Yun et al., 2020a;b) study the expressivity of Transformers from a universal-approximation perspective, by constructing networks with an exponential number of layers and parameters in input dimensions (context size and dimension) to cover the entire input space. In contrast, our aim is to find a small network to memorize a *finite* number of samples. Bhojanapalli et al. (2020) studies the expressivity of MHA on a single input sequence, yielding guidelines for setting the head dimension $d_h$ at least as high as the context size $n$ to preserve expreissivity. To the best of our knowledge, our work is the first to explicitly address memorization of MHA with more than one input data points. Only very recently, Kim et al. (2023) has extended previous results for ReLU networks (Vardi et al., 2022) to Transformers with ReLU-activation feed-forward layer and a *single* Attention head, showing that a sublinear number of parameters suffice for memorization. Here, in order to better understand the distinct role of the Attention mechanism, we study the memorization of one-layer MHA. Also, unlike those previous works, we focus on a single prediction per input. These make our results not directly comparable to theirs. Also, our assumptions on the input data and proof techniques are different.

**Memory capacity of FCNs.** Contrary to Transformers, the memory capacity of FCNs with ReLU/Threshold/Sigmoid activation has been extensively analyzed. Baum (1988) provided the first memory bound, proving under General Position that a single hidden-layer Threshold can memorize $T$ binary labels with $O(T)$ parameters. The best known result on memorization under ReLU activation and real-valued inputs was obtained only very recently by Bubeck et al. (2020). These

bounds are tight for the respective class of assumptions since memorization for a dense class of input data (*e.g.*, General Position) with piece-wise analytic definable functions requires number of parameters at least linear on the number of examples (Sontag, 1997). Building on these foundational works, recent work have extended these results to different input-data assumptions, smooth activations, and multi-layer networks (Vershynin, 2020; Yun et al., 2019; Rajput et al., 2021; Zhang et al., 2021; Bombari et al., 2022). Additionally, recent works yield bounds on ReLU networks with $\tilde{O}(\sqrt{T})$ parameters by utilizing $\tilde{\Omega}(\sqrt{T})$-deep networks (Vardi et al., 2022; Park et al., 2021). Following the tradition of early studies on FCNs that explored memorization in simplified one-hidden layer networks, we delve into the memory capacity of a single-layer MHA mechanism. Our objective is to isolate the MHA from the ReLU MLP and analyze its unique properties. We anticipate that our discoveries will inspire additional research and extensions for more advanced Transformer models.

## 3 PROBLEM SETUP

**Notation.** $[n]$ denotes the set of positive integers $\{1, 2, \ldots, n\}$. Boldface upper-case \ lower-case letters, such as $\boldsymbol{A} \setminus \boldsymbol{v}$, denote matrices \ column-vectors. $[\boldsymbol{A}; \boldsymbol{B}] \setminus [\boldsymbol{A}, \boldsymbol{B}]$ denote row-wise \ column-wise concatenation. $\boldsymbol{x}_{[n]} = \{\boldsymbol{x}_1, \boldsymbol{x}_2, \ldots, \boldsymbol{x}_n\}$ denotes the set of $[n]$ indexed vectors.

We study a single-layer *multi-head Attention* (MHA) mechanism. An MHA layer $\mathcal{A}$ with $H$ heads consists of three sets of matrices $\boldsymbol{W}_{Kh}, \boldsymbol{W}_{Qh} \in \mathbb{R}^{d \times d_h}, \boldsymbol{W}_{Vh} \in \mathbb{R}^{d \times d_v}$ for each head $h \in [H]$, one weight matrix $\boldsymbol{W}_O \in \mathbb{R}^{Hd_v \times d}$ to combine the outputs of different heads, and finally, a read-out weight matrix $\boldsymbol{W}_D \in \mathbb{R}^{d \times d_{\text{out}}}$ to get the final model's output. We denote the entire set of parameters as $\mathcal{W} = \{\{\boldsymbol{W}_{Qh}, \boldsymbol{W}_{Kh}, \boldsymbol{W}_{Vh}\}_{h=1}^H, \boldsymbol{W}_O, \boldsymbol{W}_D\}$. An input example to the MHA model consists of (i) a key matrix $\boldsymbol{E} = [\boldsymbol{e}_1^\top; \boldsymbol{e}_2^\top; \ldots; \boldsymbol{e}_n^\top] \in \mathbb{R}^{n \times d}$ containing $n$ tokens, and, (ii) a query vector $\boldsymbol{e} \in \mathbb{R}^d$. The output of the model as a function of the input example and of the set of parameters is given according to the following computational mechanism:

$$\boldsymbol{\alpha}_h := \boldsymbol{E}\boldsymbol{W}_{Kh}\boldsymbol{W}_{Qh}^\top\boldsymbol{e} \quad (\boldsymbol{\alpha}_h \in \mathbb{R}^n) \quad (1) \qquad \boldsymbol{p}_h := \boldsymbol{W}_{Vh}^\top\boldsymbol{z}_h \qquad\qquad (\boldsymbol{p}_h \in \mathbb{R}^{d_v}) \quad (4)$$

$$\boldsymbol{\theta}_h := \text{Softmax}(\boldsymbol{\alpha}_h) \quad (\boldsymbol{\theta}_h \in \mathbb{R}^n) \quad (2) \qquad \boldsymbol{o} := \boldsymbol{W}_O^\top[\boldsymbol{p}_1; \boldsymbol{p}_2; \ldots; \boldsymbol{p}_H] \qquad (\boldsymbol{o} \in \mathbb{R}^d) \quad (5)$$

$$\boldsymbol{z}_h := \boldsymbol{E}^\top\boldsymbol{\theta}_h \qquad (\boldsymbol{z}_h \in \mathbb{R}^d) \quad (3) \qquad \hat{\boldsymbol{y}} := \boldsymbol{W}_D^\top\boldsymbol{o} \qquad\qquad (\boldsymbol{y} \in \mathbb{R}^{d_{\text{out}}}). \quad (6)$$

Eqs. (1)–(4) hold for all heads $h \in [H]$, $\boldsymbol{o}$ is the output of the MHA layer, and, $\hat{\boldsymbol{y}}$ is the final output of the model. In Transformer terminology, the key matrix $\boldsymbol{E}$ is also called the *context*. Moreover, when $\boldsymbol{e} \in \boldsymbol{e}_{[n]}$, the MHA layer is called *Self-Attention*, and when the query vector is not among key vectors, the MHA layer is a particular form of *Cross-Attention*. Here, we impose *no* restrictions on the specific type of MHA as long as certain assumptions on the input data are satisfied (see Sec. 4).

We consider a training set $\mathcal{T}$ consisting of $T$ input examples. To each example $t \in [T]$, consisting of $n$ context tokens and one query token, corresponds a label $\boldsymbol{y}^{(t)}$. Thus, we have $\mathcal{T} := \{(\boldsymbol{E}^{(t)}, \boldsymbol{e}^{(t)}, \boldsymbol{y}^{(t)})\}_{t=1}^T$. Our goal is to find a parameter set $\mathcal{W}$ such that the MHA model is able to memorize any label set given the key matrices and query vectors, *i.e.*, $\boldsymbol{y}^{(t)} = f_{\mathcal{W}}(\boldsymbol{E}^{(t)}, \boldsymbol{e}^{(t)})$ for all $t \in [T]$. The maximum number $T$ of examples for which we are able to achieve this serves as a lower bound on the memorization capacity of the model.

To illustrate the role of an input example distinct components (key matrix, query vector, label), consider image classification using (say) Vision Transformer (ViT) (Dosovitskiy et al., 2021). Two types of tokens are given as input to this model: A single "[CLS]" token, and several "image patch" tokens. The image patches contain information about the input image, while the "[CLS]" token is a special token in that the model's class output is read from the prediction of this token. For each image $t \in [T]$, the key matrix $\boldsymbol{E}^{(t)}$ is the matrix containing image patches and the "[CLS]" token, the query vector $\boldsymbol{e}^{(t)}$ corresponds to the "[CLS]" token only, and $\boldsymbol{y}^{(t)}$ is the label of the image (here, the label is a scalar; thus, $d_{\text{out}} = 1$.) Note that, the type of MHA is here Self-Attention since the "[CLS]" token is inside $\boldsymbol{E}^{(t)}$ as well, otherwise, the context would have contained only the image patches.

Often, a positional encoding is also added to the embedding of each token before feeding them to the Transformer. Namely, for all $t \in [T]$: $\boldsymbol{e}^{(t)} := \boldsymbol{x}^{(t)} + \text{pos}_0$, and $\boldsymbol{e}_i^{(t)} := \boldsymbol{x}_i^{(t)} + \text{pos}_i$, for all $i \in [n]$, where $\boldsymbol{x}_i^{(t)}$ and $\boldsymbol{x}^{(t)}$ are the raw embedding of input tokens, and $\text{pos}_i \in \mathbb{R}^d$ are fixed and unique positional encoding vectors for each position index. In the case of Self-Attention, the query vector takes only one positional encoding since it is already present among $\boldsymbol{e}_{[n]}$. We will show in Sec. 4.4 that the use of positional encoding motivates our Assumption 2.

## 4 MAIN RESULTS

Two types of assumptions are typical in prior works on memorization: (1) norm-based (Vershynin, 2020; Park et al., 2021; Kim et al., 2023), and, (2) linear-independence (Baum, 1988; Bubeck et al., 2020) assumptions. The former assumes well-separated and/or bounded-norm input data, while the latter only requires some notion of linear independence among the data points. While norm-based assumptions are well-suited for discrete data such as language, linear independence is usually better suited in continuous data such as images, or continuous distributions, *e.g.* Gaussian and Uniform.

In this paper, we focus on the second type of assumptions. Thus, it is useful to recall the notions of Kruskal Rank and General Position.

**Definition 1** (Kruskal Rank). *Kruskal Rank of a set of vectors is the largest number $i$ such that every subset of size $i$ is linearly independent.*

**Definition 2** (General Position). *A set of vectors $\{x_1, x_2, \ldots, x_m\} \subset \mathbb{R}^d$ is in General Position if it has the maximal possible Kruskal Rank of $d$.*

In contrast to previous works on the memorization capacity of FCNs, we depart from the commonly used General Position assumptions and tailor our assumptions to transformers. Specifically, we make the following assumptions.

**Assumption 1.** *The set of all query vectors $\{e^{(t)} | e^{(t)} \in \mathbb{R}^d\}_{t=1}^T$ has Kruskal Rank at least $n$.*

**Assumption 2.** *For each example $t \in [T]$, the context matrix $E^{(t)} \in \mathbb{R}^{n \times d}$ has rank $n$.*

These assumptions find their motivation in experiments conducted on real data, which reveal scenarios where these assumptions hold, while the General Position assumption consistently fails (see Sec. 5.1).

We are now ready to state our main result below.

**Theorem 1.** *Consider a multi-head attention layer $\mathcal{A}$ with $H$ heads, embedding dimensions $d$, $d_v \geq d_{out} \geq 1$, and $d_h \geq 1$. Let $\mathcal{T} = \left\{ \left( E^{(t)}, e^{(t)}, y^{(t)} \right) \right\}_{t=1}^T$ be a training set with context size $n < d$. Define $r := \min(n, d_h)$. If Assumptions 1 and 2 hold, and $T \leq H (r - 1) + 1$, then, there exists a set of parameters $\mathcal{W}$ such that $\mathcal{A}$ can memorize $\mathcal{T}$.*

Theorem 1 shows that an MHA with $2Hdd_h + Hdd_v + dd_{out}$ parameters can memorize at least $H (\min(n, d_h) - 1) + 1$ examples under Assumptions 2 and 1. To gain further insights on the implications of this theorem consider three concrete settings: (1) Suppose $d_h = d$ and scalar model output $d_v = d_{out} = 1$, then a MHA with $\Theta(Hd^2)$ can memorize $\Omega(Hn)$ examples. (2) Suppose $d_h = n < d, d_v = d_{out} = 1$ then a MHA with $\Theta(Hdn)$ can memorize $\Omega(Hn)$ examples. (3) Suppose $d_h = d_v = d$ and $d$-dimensional model output $d_{out} = d$, then a MHA with $\Theta(Hd^2)$ can memorize $\Theta(Hn)$ examples with $d$-dimensional label each. The third scenario is particularly interesting. To see this further, consider the typical setting where $n$ and $d$ are of the same order ($n = \Theta(d)$), then the established bound becomes optimal. This optimality is reflected in the fact that $\Omega(Hnd) = \Omega(Hd^2)$ real-valued outputs are memorized, and a trivial upper bound for the memorization of an MHA is the number of its parameters, which is $\Theta(Hd^2)$[2].

Additionally, Theorem 1 suggests there is *no* gain in memorization capacity provided $d_h > n$. This agrees with using $d_h < d$ in practice. We also confirm this in synthetic experiments in Section 5.2. A similar conclusion is also drawn by Bhojanapalli et al. (2020), where authors show by having $d_h < n$, Attention loses expressivity in terms of convex combinations $\theta_h$ the Attention can represent. However, their finding alone does not allow direct conclusions about the impact on memorization. Our theorem directly connects the trade-off between $n$ and $d_h$ to memorization.

The assumption that $n < d$ is general practice. For instance, in prominent models such as Vision Transformer (Dosovitskiy et al., 2021), BERT (Devlin et al., 2019), and GPT-3 (Brown et al., 2020), the $n/d$ ratio is $197/768$, $512/768$, and $2048/12288$, respectively. Furthermore, in Sections 4.4 and 5.1, we demonstrate that Assumption 2 typically holds in practice due to positional encoding, and Assumption 1 is valid after applying a single Attention layer. Unlike previous works on FCNs that regard General Position as an assumption (Baum, 1988; Bubeck et al., 2020), our approach innovatively incorporates Assumption 1, which imposes a weaker constraint than General Position

---

[2] $d$-dimensional outputs is common in prior works, *e.g.*, (Yun et al., 2020b;a), with the distinction that they consider sequence-to-sequence outputs.

since $n < d$. In fact, our experiments reveal that Assumption 1 is satisfied in real data, even though the more commonly used General Position assumption is often not met.

## 4.1 PROOF OF THEOREM 1

Our proof consists of two steps: (1) We focus on the intermediate representations originating from $z_h$ in Equation (3). We demonstrate with a suitable set of attention weights $\{(W_{Kh}, W_{Qh})\}_{h=1}^{H}$, we can achieve linear independence across different data points, resulting in a rich representation. (2) We prove the existence of $\{\{W_{Vh}\}_{h=1}^{H}, W_O, W_D\}$ for achieving memorization by taking advantage of the rich representation from the previous step.

Let us define matrix $Z$, which is a function input data and $W_{Kh}, W_{Qh}, h \in [H]$ as follows:

$$Z := \left[ z^{(1)^\top}; z^{(2)^\top}; \ldots; z^{(T)^\top} \right] \in \mathbb{R}^{T \times dH}, \text{ where } z^{(t)} := \left[ z_1^{(t)^\top}, z_2^{(t)^\top}, \ldots, z_H^{(t)^\top} \right]^\top. \quad (7)$$

Each vector $z^{(t)}$ is a concatenation of $H$ $d$-dimensional vectors each representing a convex combination of context vectors of the $t$-th example parameterized in terms of key/query vectors and $W_{Kh}, W_{Qh}$. The matrix $Z$ gathers all these $dH$ dimensional representations, one for each example in a matrix form. Then, based on Eq. (1) to Eq. (6), the predicted labels $\hat{y}^{(t)}$ are given by

$$\hat{y}^{(t)} := \underbrace{Z}_{\text{Step 1}} \underbrace{\text{diag}(W_{V1}, W_{V2}, \ldots, W_{VH}) W_O W_D}_{\text{Step 2}} \in \mathbb{R}, \quad \forall t \in [T]. \quad (8)$$

In the first step, using an inductive proof technique, we show that $Z$ has a high rank, and use this information in the second step to solve a linear system of equations and achieve memorization. *Step 1. The Rank of $Z$.* Our key technical result is proving the following lower bound on $\text{rank}(Z)$.

**Proposition 1.** *Under the conditions of Theorem 1, there exists parameters $\{(W_{Kh}, W_{Qh})\}_{h=1}^{H}$ such that $\text{rank}(Z) \geq \min\{H(r-1) + 1, T\}$.*

We provide a proof sketch of this proposition here and defer the full proof to Appendix B. The high-level idea is to inductively add heads and use each new head for a distinct set of $r - 1$ examples, which in turn translates to an increase on the rank of $Z$. The new head exploits a new distinct set of examples without affecting the rank of the previous induction step. To accomplish this, we leverage the saturation property of the softmax and the linear independence assumptions. For simplicity, we only consider here the case $d_h \geq d > n$, for which $r = n$.

In order to get $H(n-1) + 1$ rank, each time a head is added, the rank of $Z$ should be increased by $n - 1$. For each head $h$, we only focus on $h(n-1) + 1$ examples, therefore, the subsequent step can be viewed as adding a new head with tunable weights, and a set of $n - 1$ fresh data points. Namely, the induction step of $Z$ can be described in view of the following block matrix:

$$Z = \left[ \begin{array}{c|c} Z_a' & Z_a'' \\ \hline Z_b' & Z_b'' \end{array} \right] \xrightarrow{\text{row-operations}} \left[ \begin{array}{c|c} Z_a' & Z_a'' \\ \hline \mathbf{0} & Z_b'' + \mathbf{A} \end{array} \right] \begin{array}{l} \left.\rule{0pt}{12pt}\right\} T':=T-n+1 \\ \left.\rule{0pt}{12pt}\right\} n-1 \end{array}$$
$$\underbrace{\phantom{Z_a'}}_{H'd} \underbrace{\phantom{Z_a''}}_{d} \qquad \underbrace{\phantom{Z_a'}}_{H'd} \underbrace{\phantom{Z_a''}}_{d}$$

$Z_a' \in \mathbb{R}^{T' \times H'd}$ denotes the matrix derived from the induction hypothesis, $Z_b' \in \mathbb{R}^{(n-1) \times H'd}$ contains the embeddings of new examples corresponding to the $H' \geq 1$ heads of the previous induction steps, $Z_a'' \in \mathbb{R}^{T' \times d}$ contains the embeddings of the previous examples corresponding to the new head $H = H' + 1$, and finally, $Z_b'' \in \mathbb{R}^{(n-1) \times d}$ contains the embeddings of the new examples corresponding to the new head $H$. The upper-triangular block matrix on the right-hand side is obtained from $Z$ only by row operations (see the appendix for details); thus, (i) its rank is the same as the rank of $Z$, and, (ii) the matrix $\mathbf{A} := \mathbf{A}(Z_a'')$ is a function of the block $Z_a''$.

In view of the upper-triangular block formulation above, our objective becomes clear.

**Goal:** Adjust the weights $W_{KH}, W_{QH}$ of the new head $H$, to increase the rank of $Z_b'' + \mathbf{A}$ while dealing with the challenge that such tuning can also impact $Z_a''$, subsequently affecting the rank of $\mathbf{A} = \mathbf{A}(Z_a'')$. We will demonstrate a method for precisely tuning these weights to control the rank of $Z_b''$ without altering the rank of $Z_a''$.

**Approach:** We approach this in two sub-steps, providing an overview of each below. Recall $\boldsymbol{\theta}_H^{(t)}(\mathbf{W}) := \text{Softmax}\left(\mathbf{E}^{(t)}\mathbf{W}\mathbf{e}^{(t)}\right)$ is the vector of softmax coefficients, associated with the $t$-th input and the new head $H$, parameterized by $\mathbf{W} := \boldsymbol{W}_{KH}\boldsymbol{W}_{QH}^\top$. Also note that $\boldsymbol{Z}_a''$ and $\boldsymbol{Z}_b''$ are parameterized by $\boldsymbol{\theta}_H^{(t)}(\mathbf{W})$. Consequently, we will use $\boldsymbol{Z}_a''(\mathbf{W})$ and $\boldsymbol{Z}_b''(\mathbf{W})$ to explicitly denote parameterization with a specific weight matrix.

● *Sub-step 1.A. Tune the new head's weights and memorize the new set of examples.* Given an arbitrary, but fixed matrix $\boldsymbol{M} \in \mathbb{R}^{(n-1)\times d}$, whose precise value will be chosen later in the second substep, we demonstrate that it is possible to design weights $\mathbf{W}^* := \mathbf{W}^*(\boldsymbol{M})$ to achieve a target rank for $\boldsymbol{Z}_b''$, specifically to achieve $\text{rank}(\boldsymbol{Z}_b''(\mathbf{W}^*) + \boldsymbol{M}) \geq n-1$. The construction is based on an $n$-step induction, making use of Assumption 2 that the rows of $\mathbf{E}^{(t)}$ span an $n$-dimensional space.

● *Sub-step 1.B. Suppress the impact of the new head's weights on the previous examples.* To select the matrix $\boldsymbol{M}$, we establish the following procedure for fixing $\boldsymbol{Z}_a''$. We first establish (see Lemma 3) the existence of a matrix $\mathbf{W}^+$, satisfying two conditions: (i) $\mathbf{E}^{(t)}\mathbf{W}^+\mathbf{e}^{(t)} = 0$ for all $t = T'+1, \ldots, T$, and (ii) $\mathbf{e}_i^{(t)}\mathbf{W}^+\mathbf{e}^{(t)} \neq \mathbf{e}_j^{(t)}\mathbf{W}^+\mathbf{e}^{(t)}$ for all $i \neq j \in [n]$ and $t \in [T']$. Given $\mathbf{W}^+$, define $\boldsymbol{\theta}_+^{(t)} := \mathbf{1}[\arg\max_{i\in[n]} \mathbf{e}_i^{(t)}\mathbf{W}^+\mathbf{e}^{(t)}]$ for $t \in [T']$. The crucial insight here is that owing to the saturating property of the softmax (see Lemma 4) and property (ii) above, for *any* weight matrix $\mathbf{W}$ (which may depend on $\mathbf{W}^+$), we have $\boldsymbol{\theta}_+^{(t)} = \lim_{c\to\infty} \boldsymbol{\theta}_H^{(t)}(\mathbf{W} + c\mathbf{W}^+)$ for all $t \in [T']$. Consequently, for *any* $\mathbf{W}$, the matrix $\lim_{c\to\infty} \boldsymbol{Z}_a''(\mathbf{W} + c\mathbf{W}^+) = [\boldsymbol{E}^{(1)\top}\boldsymbol{\theta}_+^{(1)} \cdots \boldsymbol{E}^{(T')\top}\boldsymbol{\theta}_+^{(T')}]^\top =: \boldsymbol{Z}_{a,+}$ is independent of the choice of $\mathbf{W}$.

The proof is finalized by integrating the two sub-steps. Specifically, we utilize $\boldsymbol{Z}_{a,+}$ from sub-step 1.B and select $\boldsymbol{M} = \mathbf{A}(\boldsymbol{Z}_{a,+})$. Next, we apply sub-step 1.A to identify the weights $\mathbf{W}^* = \mathbf{W}^*(\mathbf{W}^+)$ in a manner that ensures $\text{rank}(\boldsymbol{Z}_b''(\mathbf{W}^*) + \mathbf{A}(\boldsymbol{Z}_{a,+})) \geq n-1$. Finally, we set the overall weights of the new head as $\mathbf{W} = \mathbf{W}^* + c\mathbf{W}^+$. As the parameter $c$ approaches infinity, the softmax coefficients for the first $T'$ examples will be exclusively determined by $\mathbf{W}^+$ (not $\mathbf{W}^*$), as observed in Sub-step 1.B. Furthermore, thanks to property (i) of Sub-step 1.B, we have $\boldsymbol{Z}_b''(\mathbf{W}^* + c\mathbf{W}^+) = \boldsymbol{Z}_b''(\mathbf{W}^*)$ for all values of $c$. Collectively, these steps complete the proof of the proposition.

In the Appendix B, we provide all the details. This involves showing that a finite value of scaling $c$ above suffices; thus, there exist *finite* weights with the desired property. Moreover, for the scenario $d_h \leq d$, we demonstrate that the matrices $\mathbf{W}^*, \mathbf{W}^+$ can be chosen to have rank at most $d_h$.

*Step 2. Solving system of equations.* We can find the remaining weights to memorize at least $\text{rank}(\boldsymbol{Z})$ examples. WLOG, assume that $T = \text{rank}(\boldsymbol{Z})$, otherwise we can ignore some of the data points and keep only $\text{rank}(\boldsymbol{Z})$ data points that make $\boldsymbol{Z}$ full-rank. We find weights $\{\{\boldsymbol{W}_{Vh}\}_{h=1}^H, \boldsymbol{W}_O, \boldsymbol{W}_D\}$ to memorize all $T$ examples as follows:

$$\boldsymbol{Z}\boldsymbol{W}_V = \left[\begin{array}{c|c} \begin{array}{c} \boldsymbol{y}^{(1)\top} \\ \vdots \\ \boldsymbol{y}^{(T)\top} \end{array} & \boldsymbol{0}_{T\times(d_v-d_{out})} \end{array}\right], \quad \begin{bmatrix} \boldsymbol{W}_{V1} \\ \boldsymbol{W}_{V2} \\ \vdots \\ \boldsymbol{W}_{VH} \end{bmatrix} := \boldsymbol{W}_V, \quad \boldsymbol{W}_O := \begin{bmatrix} \boldsymbol{I}_{d_v\times d} \\ \boldsymbol{I}_{d_v\times d} \\ \vdots \\ \boldsymbol{I}_{d_v\times d} \end{bmatrix}, \quad \boldsymbol{W}_D := \boldsymbol{I}_{d\times d_{out}},$$

where $\boldsymbol{W}_V$ is a solution to the first expression (note that such $\boldsymbol{W}_V$ exists since $\boldsymbol{Z}$ has full row rank). Substituting in Eq. (8), this gives $\hat{\boldsymbol{y}}^{(t)} = \boldsymbol{y}^{(t)}$ for all $t \in [T]$. Therefore, we have proved that the memorization capacity of one layer is at least $\text{rank}(\boldsymbol{Z})$. But, since we also proved a lower bound of $\text{rank}(\boldsymbol{Z}) \geq H(r-1) + 1$, we conclude that the memorization capacity is at least $H(r-1) + 1$.

**Remark 1.** *In the setting of Theorem 1, if Assumption 1 is violated (i.e., the Kruskal Rank of query vectors is less than $n$), define $Q < n$ to be the Kruskal Rank of query vectors $\{\boldsymbol{e}^{(t)}|\boldsymbol{e}^{(t)} \in \mathbb{R}^d\}_{t=1}^T$. Then, a weaker lower bound on the memorization still holds: $T \leq H(\min(Q, d_h) - 1) + 1$.*

## 4.2 Tightness of the lower bound on the rank

Our proof of Theorem 1 relies on lower bounding $\text{rank}(\boldsymbol{Z})$ in Proposition 1. One might naturally wonder how tight this bound is. The proposition below proves that the bound is indeed tight.

**Proposition 2.** *Under the conditions of Theorem 1, if all the contexts are shared, i.e., $\boldsymbol{E}^{(t)} := \boldsymbol{E}$ for all $t \in [T]$, then for any $\{(\boldsymbol{W}_{Kh}, \boldsymbol{W}_{Qh})\}_{h=1}^H$, we have $\text{rank}(\boldsymbol{Z}) \leq H(n-1) + 1$.*

We defer the proof to Appendix E. The proposition shows that the lower bound of Proposition 1 is tight when the context is shared across data points and $d_h \geq n$. Furthermore, the setting of Proposition 2 better highlights the dependency factor of $r - 1$ rather than $r$. To illustrate this, consider this setting for $r = n = 1$, *i.e.*, each context contains a single token and is shared across all examples. Then, all $\boldsymbol{\theta}_h^{(i)}$ become scalars, whose values are equal to one. Consequently, the vector $\boldsymbol{z}_h^{(t)}$ copies the first (and only) context token, which is shared across all examples. Hence, no matter how large $H$ or $d$ are, the output of MHA for all examples is identical, and the memory capacity of the attention is exactly $T = H(r - 1) + 1 = 1$ (*i.e.*, a single example).

### 4.3 COMPARISON WITH TWO-LAYER ReLU NETWORKS

Given Assumptions 1 and 2, we compare the memorization capacity of MHA with one-hidden layer FCN networks of the same size. An FCN does not distinguish between key/query vectors. Hence, we flatten out the concatenations of $\boldsymbol{E}^{(t)}$ and $\mathbf{e}^{(t)}$, resulting in input $\boldsymbol{x}_{\text{FCN}}^{(t)} := [\boldsymbol{e}_1^{(t)}; \ldots; \boldsymbol{e}_n^{(t)}; \boldsymbol{e}^{(t)}] \in \mathbb{R}^{(n+1)d}, t \in [T]$. The proposition below upper bounds the memorization power of a ReLU FCN.

**Proposition 3.** *Suppose a two-layer ReLU network with input dimension $(n+1)d$, hidden dimension $m$, and output dimension $d_{out}$. Then, the memorization power of this network for the class of datasets $\mathcal{T} = \{(\boldsymbol{E}^{(t)}, \boldsymbol{e}^{(t)}, \boldsymbol{y}^{(t)})\}_{t=1}^T$ with context size $n < d$ and satisfying Assumptions 1 and 2 is at most $T \leq (n+1)m/d_{out} + (m+1)$ examples.*

We defer the proof to Appendix F. A ReLU FCN with $\Theta(ndm + md_{out})$ parameters can memorize at most $O(nm/d_{out} + m)$ examples under Assumptions 1 and 2. This setting is particularly interesting with $m = \Theta(H)$ and $d_{out} = 1$, where a two-layer ReLU network with $\Theta(Hnd)$ can memorize at most $O(Hn)$ examples. This *upper* bound on memorization of a ReLU network together with the *lower* bound of $\Omega(Hn)$ as established in Theorem 1 for MHA with the same order of parameters, shows that the latter is at least as powerful, in terms of memorization, as the former (when data satisfy Assumptions 1 and 2). This also aligns with practical Transformer implementations, where a similar order of parameters is allocated to both Attention and ReLU FCN (*e.g.*, (Touvron et al., 2023).

### 4.4 ANALYSIS OF THE VALIDITY OF ASSUMPTIONS

We confirm our assumptions are valid either on the embedding layer or after just one Attention layer. Detailed experimental verification of this full-rank assumption is provided in Section 5.1.

First, we motivate Assumption 2 by considering the common practice of incorporating positional encoding into token representations before inputting data into a Transformer. A widely used positional encoding method utilizes sinusoidal positional encodings (Vaswani et al., 2017). Consequently, the positional encoding matrix is inherently full rank. In view of the positional encoding equations (see Sec. 3), this characteristic extends to the context matrix $\boldsymbol{E}^{(t)}$, establishing the validity of Asm. 2.

Second, we provide an explanation for why Assumption 1 generally becomes valid after just one Attention layer. Consider the case of image classification using ViT, where query vectors $\boldsymbol{e}^{(t)}$ are all composed of the "[CLS]" token combined with positional encoding at position zero. Here, Assumption 1 does not initially hold. However, we demonstrate below that it becomes valid after a single Self-Attention layer with a skip connection. The key insight lies in the behavior of the "[CLS]" token: as it traverses the attention layer, it effectively mixes information from the surrounding context, which is distinct for each image. Therefore, after a single (even randomly initialized and fixed) Self-Attention layer, we establish that Assumptions 2 and 1 are satisfied. This allows us to directly apply Theorem 1 to the second layer of Attention to achieve memorization. This line of argument is conceptually similar to the approach employed by Vershynin (2020), who first designs layers with random weights to enrich the representation of data points, and then proves the existence of a learned subsequent perception layer for memorizing the labels in a FCN.

Now, we provide the details on how one Attention-layer with skip connection validates Assumption 1. To begin with, we recall how a Self-Attention layer with a skip connection works. Consider a single example with context matrix $\boldsymbol{E}$ and query token $\boldsymbol{e}$. Given a set of attention weights $\mathcal{W}$, the Self-Attention layer with skip connection transforms the tokens according to the following equations:

$$\boldsymbol{e}' := \boldsymbol{e} + f_{\mathcal{W}}(\boldsymbol{E}, \boldsymbol{e}), \quad \boldsymbol{e}_i' := \boldsymbol{e}_i + f_{\mathcal{W}}(\boldsymbol{E}, \boldsymbol{e}_i) \quad \text{for all } i \in [n], \quad \boldsymbol{E}' := [\boldsymbol{e}_1'^{\top}; \boldsymbol{e}_2'^{\top}; \ldots; \boldsymbol{e}_n'^{\top}],$$

where $\boldsymbol{e}'$, and $\boldsymbol{E}'$ correspond to the output of query, and context vectors, respectively. Note that the query matrix is part of the context matrix due to Self-Attention. Let us assume the query vector is in the first row of $\boldsymbol{E}$, *i.e.* $\boldsymbol{e}_1 = \boldsymbol{e}$, and consequently $\boldsymbol{e}' = \boldsymbol{e}_1'$.

**Remark 2.** *Theorem 1 also holds true for Attention layers with skip connection (see Appendix C).*

We are now ready to state a proposition on the mixing power of this type of Attention with skip connection. To do this, we introduce the following relaxed assumption.

**Assumption 3.** *For training set $\mathcal{T} = \left\{ \left( \boldsymbol{E}^{(t)}, \boldsymbol{e}^{(t)}, \boldsymbol{y}^{(t)} \right) \right\}_{t=1}^{T}$, define $\tilde{\boldsymbol{e}}^{(t)} := \boldsymbol{e}^{(t)} + \frac{1}{n} \sum_{i \in [n]} \boldsymbol{e}_i^{(t)}$ for all $t \in [T]$. Then the set $\mathcal{S} := \left\{ \tilde{\boldsymbol{e}}^{(1)}, \tilde{\boldsymbol{e}}^{(2)}, \ldots, \tilde{\boldsymbol{e}}^{(T)} \right\}$ has Kruskal Rank at least $n$.*

**Proposition 4.** *Define Self-Attention layer $\mathcal{A}_0$ with single head, skip connection, $d_v = d$, and weights $\boldsymbol{W}_K = \boldsymbol{W}_Q = \boldsymbol{0}, \boldsymbol{W}_V = \boldsymbol{W}_O = \boldsymbol{I}$. Suppose training set $\mathcal{T} = \left\{ \left( \boldsymbol{E}^{(t)}, \boldsymbol{e}^{(t)}, \boldsymbol{y}^{(t)} \right) \right\}_{t=1}^{T}$ satisfies Assumptions 2 and 3. Let $\boldsymbol{E}'^{(t)}, \boldsymbol{e}'^{(t)}$ be the output of $\mathcal{A}_0$ for each example $t \in [T]$ and define new dataset $\mathcal{T}' := \left\{ \left( \boldsymbol{E}'^{(t)}, \boldsymbol{e}'^{(t)}, \boldsymbol{y}^{(t)} \right) \right\}_{t=1}^{T}$. Then, $\mathcal{T}'$ satisfies both Assumptions 2 and 1.*

Assumption 3 assumes that the input data as a whole is sufficiently distinctive across different examples. In particular, this is a more relaxed assumption compared to Assumption 2. For instance, in the case of images, it corresponds to the average of all image patches (plus a constant vector) being sufficiently distinctive across different images. Proposition 4 establishes that under the given assumptions (specifically, Asm. 3 in conjunction with Asm. 2), the application of a trivial Self-Attention layer results in the fulfillment of Assumption 1 at the output of the Attention layer, all while preserving Assumption 2. This implies that the "[CLS]" token is capable of performing token mixing even with a basic Attention mechanism. Consequently, Theorem 1 can be directly applied to a second layer of Attention, facilitating effective memorization of the examples.

## 5 EXPERIMENTS

### 5.1 TESTING ASSUMPTIONS ON VISION TRANSFORMER (ViT)

As mentioned in Section 4.4, we show that our assumptions on the input tokens hold after applying a single Attention layer. To test this, we mainly focus on image classification and Vision Transformer (ViT) (Dosovitskiy et al., 2021; Wolf et al., 2020). We consider the following models: 1) **Embedding Layer:** A randomly initialized embedding layer. This layer contains only positional encoding and a linear embedding layer (*i.e.*, the input tokens $\boldsymbol{E}^{(t)}, \boldsymbol{e}^{(t)}$). 2) **Random Attention:** A randomly initialized Self-Attention layer with similar architecture to a ViT layer, without the FCN of a Transformer. 3) **Random ViT:** A randomly initialized base ViT. We only look at the output of the first layer and discard the subsequent layers. 4) **Trained ViT:** Similar to the random ViT, with the difference that we take the weights from the ViT pre-trained on ImageNet (Deng et al., 2009).

We evaluate the mentioned models on 2000 images sampled from ImageNet. To empirically test Assumption 2, we verify whether the context vectors are all linearly independent for each example. On the other hand, testing Assumption 1 is computationally difficult since computing Kruskal Rank is NP-Hard. Therefore, it can only be approximately tested in polynomial time. To address this, we randomly sample $n$ query vectors from the training set $\mathcal{T}$ and construct a matrix of size $n \times d$. We then test if the rank of the matrix is equal to $n$. After repeating this test 5000 times, we report that the assumption holds if the rank check holds for at least $99\%$ of the test instances. The results of our empirical tests are summarized in Table 1. Specifically, we find that only Assumption 1 is violated in the Embedding layer, whereas both assumptions hold true for the remaining models. Notably, the commonly made assumption of General Position fails in all cases, and the Kruskal Rank is typically much less than $d$, as shown by Figure 1. We also empirically test Assumption 3 and find that it already holds for the Embedding layer. Following up on Section 4.4, this may explain why Assumption 1 already holds after applying any type of Attention. [3]

### 5.2 EMPIRICAL VALIDATION OF MEMORIZATION IMPROVEMENT FACTORS

Theorem 1 provides three conclusions: (1) When fixing dimension $d$, increasing the number of heads $H$ improves memorization. (2) When further fixing the number of heads $H$, increasing the context size $n$ improves memorization. (3) When fixing $d, n$, increasing $d_h$ only helps up to $d_h < n$, and there is no memorization gain beyond that. By conducting synthetic experiments, we verify the conclusions drawn by the theorem. Throughout, we fix dimension $d = 64$. Moreover, to follow Assumptions 2 and 1, we generate random query inputs, and random shared context inputs (see Appendix I.1 for

---

[3]Our code is available at https://github.com/smahdavi4/attention-memorization

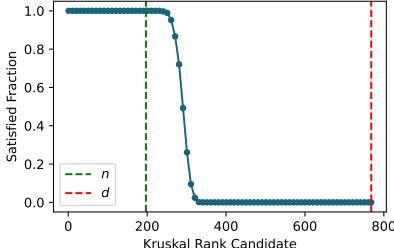

| Model | Gen. Pos. | Asm. 1 | Asm. 2 |
|-------|-----------|--------|--------|
| Embedding | ✗ | ✗ | ✓ |
| Rand Attention | ✗ | ✓ | ✓ |
| Rand ViT | ✗ | ✓ | ✓ |
| Trained ViT | ✗ | ✓ | ✓ |

Figure 1: Testing Kruskal Rank of query tokens on the output of one layer Random Attention on ImageNet. The Kruskal Rank is only slightly larger than $n$ (Assumption 1), and much smaller than $d$ (General Position).

Table 1: Testing assumptions on ImageNet and ViT: while only one assumption is violated in the embedding layer, both hold true after a single layer of Attention or Transformer. The General Position assumption is violated in all cases.

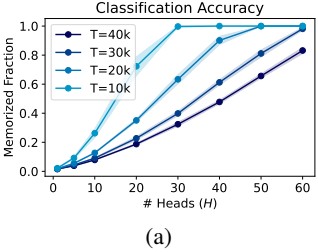
(a)

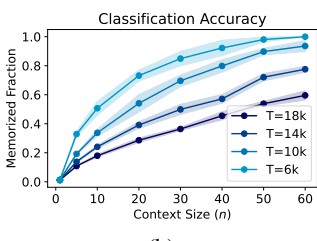
(b)

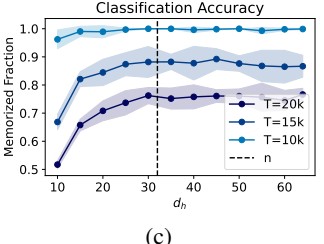
(c)

Figure 2: Testing memorization as a function of number of heads (a), context size (b), and head size (c) for classification under Assumptions 2 and 1. Examples generated synthetically with $d = 64$ and shared context (see Proposition 2). Memorization increases linearly with $H$, monotonically with $n$, and monotonically with $d_h$ as long as $d_h \leq n$.

experiments with non-shared context) according to a uniform distribution, as follows:

$$\boldsymbol{x}_1, \boldsymbol{x}_2, \ldots, \boldsymbol{x}_n \overset{iid}{\sim} \mathcal{U}(0,1)^{64}, \quad \boldsymbol{x}^{(t)} \overset{iid}{\sim} \mathcal{U}(0,1)^{64}, \quad \boldsymbol{x}_i^{(t)} := \boldsymbol{x}_i \quad \text{for all } i \in [n], \ t \in [T],$$

where $\boldsymbol{x}_i^{(t)}, \boldsymbol{x}^{(t)}$ are the raw inputs. Each input is then given to the model consisting of an embedding layer, positional encoding, and an Attention layer. To test the first conclusion, we fix $n = 32, d_h = d$ and increase $H$. To test the second conclusion, we fix $H = 20, d_h = d$ and increase context size $n$. In addition to classification tasks, we also include regression tasks in the appendix.

The results of our experiments are presented in Figure 2 for classification task with various dataset sizes $T$ and uniform labels $y \in \mathcal{U}([100])$ from 100 classes (see Figure 5 in the appendix for the regression task). We measure memorization by computing average accuracy across examples. Observe that memorization increases monotonically with $n$ and linearly with $H$. To quantify the linearity with $H$, we measure the $R^2$ score of the best linear fit of the non-saturated part in Figure 2a and observe $R^2 \geq 0.98$ in all cases, confirming a high degree of linearity. Also observe in Figure 2c that there appears no benefit on memorization fraction beyond $d_h > n$.

## 6 CONCLUSION AND DISCUSSION

In this paper, we proved a lower bound on the memorization capacity of attention layers under a rather mild linear independence assumption. Our proof technique relies on establishing a tight bound on the rank of a matrix of intermediate representations in the attention layer. We discussed implications of our lower bound which we also confirmed by additional synthetic experiments. Our work opens up several promising avenues for future research. First, it would be valuable to extend our theoretical results beyond one layer of attention networks and to sequence-to-sequence learning scenarios. Second, while we studied some specific cases of interest, establishing general upper bounds on the memorization capacity of MHA remains an important open question. Finally, refining our data assumptions to narrow the gap between practical memorization abilities and theoretical guarantees could shed more light on the memorization capacity of Attention networks (see Appendix I). Ultimately, we hope that gaining a better understanding of the memorization abilities of Transformers could facilitate more efficient implementations and better use of data for training more effective, generalizable, and privacy-preserving models.

ACKNOWLEDGMENTS

This work was funded, in part, by NSERC DG Grants (No. RGPIN-2022-04636 and No. RGPIN-2021-03677), the Vector Institute for AI, Canada CIFAR AI Chair, and Oracle Cloud credits. Resources used in preparing this research were provided, in part, by the Province of Ontario, the Government of Canada through CIFAR, and companies sponsoring the Vector Institute (`www.vectorinstitute.ai`), Advanced Research Computing at the University of British Columbia, the Digital Research Alliance of Canada (`alliancecan.ca`), and the Oracle for Research program.

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

| Symbol | Description |
|---|---|
| $\mathcal{A}$ | A single-layer multi-head Attention (MHA) mechanism |
| $\mathcal{T}$ | Training set |
| $H$ | Number of heads in the MHA layer |
| $T$ | Training set size |
| $n$ | Context length |
| $\boldsymbol{W}_{Kh}, \boldsymbol{W}_{Q_h} \in \mathbb{R}^{d \times d_{\mathrm{h}}}, \boldsymbol{W}_{Vh} \in \mathbb{R}^{d \times d_v}$ | Weight matrices for each head $h \in [H]$ |
| $\boldsymbol{W}_O \in \mathbb{R}^{Hd_v \times d}$ | Weight matrix to combine the outputs of different heads |
| $\boldsymbol{W}_D \in \mathbb{R}^d$ | Final read-out weight of the model |
| $\mathcal{W}$ | The entire set of parameters of the MHA layer |
| $\boldsymbol{E} \in \mathbb{R}^{n \times d}$ | Key/Context matrix containing $n$ key/context vectors |
| $\boldsymbol{e} \in \mathbb{R}^d$ | Query vector |
| $\boldsymbol{y} \in \mathbb{R}$ | Ground truth label |
| $\boldsymbol{\alpha}_h, \boldsymbol{\theta}_h, \boldsymbol{z}_h, \boldsymbol{p}_h$ | Intermediate variables in the MHA layer |
| $\boldsymbol{o}$ | Output of the MHA layer |
| $\hat{\boldsymbol{y}} \in \mathbb{R}$ | Final output of the model |

Table 2: Important notations used in the paper.

## A  EXPERIMENT DETAILS

In the synthetic experiment, for regression tasks, we generate uniform labels $y^{(t)} \sim \mathcal{U}(0,1)$, and for classification tasks, we draw uniform labels from 100 classes $y^{(t)} \in \mathcal{U}([100])$. For classification, we measure memorization by computing the average accuracy across examples in a dataset. For regression, we use the mean squared error (MSE) loss to compare memorization abilities. We train each task for at least 500 epochs and at least 50,000 optimization steps (whichever is larger) with a batch size of 256, using an Adam optimizer with a learning rate of 0.001, and a scheduler with linear warmup and Cosine decay. For the experiments reported in Figure 2c and Figure 3 since the optimization is more challenging, we train for a total of 300,000 optimization steps.

While our experimental results in Section 5.2 conclusively show that increasing the number of heads $H$ or context size $n$ improves the model's memorization capability, we caution the reader that the experiments are done purely through gradient descent optimization, which has limitations when it comes to studying memorization. First, it is not guaranteed that if a memorizing solution exists, GD would be able to find it. Second, optimization requires hyperparameter tuning and a large number of gradient updates to achieve memorization, especially in under-parameterized regimes and large-context sizes (Garg et al., 2022). We mitigate these issues by running small-scale experiments with a sufficiently high number of optimization steps and adequate learning rate scheduling. Running the experiments reported in our paper takes approximately 20 GPU days on an NVIDIA V100 GPU. However, this procedure becomes highly resource-exhaustive and prohibitive in larger-scale experiments. For instance, we find out that having large $d_h$ (beyond $n$) can generally improve optimization speed, and lead to better memorization. However, with sufficient large steps, this gap closes, and gains in memorization vanish in the $d_h > n$ regime (as observed in Figure 2c).

## B  PROOF OF PROPOSITION 1

For convenience let us define $r = \min(n, d_h)$. WLOG let us assume that $T \geq H(r-1) + 1$. This is because for the case of $T < H(r-1) + 1$ we can add extra dummy data points that satisfy Assumptions 1 and 2 and make $T = H(r-1) + 1$, then prove $\boldsymbol{Z}$ has full row-rank, and finally remove the extra rows of $\boldsymbol{Z}$ (*i.e.* dummy data points) to prove the statement.

We prove the lemma by induction on $H$. We will demonstrate that by adding each head, we can increase the rank of $\boldsymbol{Z}$ by $r - 1$. The key to proving the induction is the following two claims:

**Claim 1** (Fit $r$ examples in one head). *Assume that $r$ examples $\left\{ \left( \boldsymbol{E}^{(t)}, \boldsymbol{e}^{(t)} \right) \right\}_{t=1}^r$ satisfy Assumptions 1 and 2. Fix any head index $h \in [H]$. For any arbitrary set of convex combination vectors*

$\boldsymbol{\theta}_{h,*}^{(t)}\ \forall t \in [r]$, *there exists an Attention head with key/query weights* $\boldsymbol{W}_{Kh}, \boldsymbol{W}_{Qh}$ *which achieves such coefficients for the set of* $r$ *data points.*

**Claim 2** (Fit $r - 1$ examples in one head, leave the rest intact)**.** *Assume that* $T' \geq r$ *examples* $\left\{\left(\boldsymbol{E}^{(t)}, \boldsymbol{e}^{(t)}\right)\right\}_{t=1}^{T'}$ *satisfy Assumptions 1 and 2, and define* $\mathcal{R} = \{T' - r + 2, \ldots, T' - 1, T'\}$ *as the set of last* $r - 1$ *examples' indices and* $\mathcal{R}^c = [T] - \mathcal{R}$ *as its complement for convenience. Fix any head index* $h \in [H]$. *Then, there exist* $\boldsymbol{\theta}_+^{(t)}, t \in \mathcal{R}^c$ *such that for any arbitrary set of desired convex combination vectors* $\boldsymbol{\theta}_*^{(t)}, t \in \mathcal{R}$ *and any* $\delta > 0$, *there exists key/query weights* $\boldsymbol{W}_{Kh}, \boldsymbol{W}_{Qh}$ *such that both* $\boldsymbol{\theta}_h^{(t)} = \boldsymbol{\theta}_*^{(t)}, \forall t \in \mathcal{R}$ *and* $\left\|\boldsymbol{\theta}_h^{(t)} - \boldsymbol{\theta}_+^{(t)}\right\|_1 \leq \delta, t \in \mathcal{R}^c$.

Claim 1 states that we can design an attention head that produces any arbitrary $\boldsymbol{\theta}_h$ coefficients for $r$ examples. Furthermore, Claim 2 reinforces Claim 1 by demonstrating the possibility of tuning coefficients $\boldsymbol{\theta}_h$ of $r - 1$ examples while keeping the rest of the examples approximately fixed with any precision $\delta$. However, this comes at the cost of fitting one less example into a single head. The proofs of both claims will be deferred until the end of the proposition. We are now ready to prove the proposition by induction on $H$.

*Step 1. The base of induction* $H = 1$. First, we prove the base of induction for $H = 1$. Let us only consider the first $r$ examples (*i.e.*, $T' = r$). Notice that the matrix $\boldsymbol{Z}$ has $r$ rows and $d$ columns in this case. Moreover, the $t$-th row of $\boldsymbol{Z}$ has the form

$$\boldsymbol{Z}_t = \boldsymbol{E}^{(t)\top} \boldsymbol{\theta}_1^{(t)} = \left[\boldsymbol{e}_1^{(t)}, \boldsymbol{e}_2^{(t)}, \ldots, \boldsymbol{e}_n^{(t)}\right] \boldsymbol{\theta}_1^{(t)},$$

and is a convex combination of the context tokens for the example $t$. According to Assumption 2, the span of context vectors $\boldsymbol{e}_{[n]}^{(t)}$ has dimension $n$. Therefore, we can inductively find $\boldsymbol{\theta}_1^{(1)}, \boldsymbol{\theta}_1^{(2)}, \ldots, \boldsymbol{\theta}_1^{(r)}$ such that each row $\boldsymbol{Z}_t$ is not in the span of the previous $t - 1$ rows. We can continue this process for all $t \leq r$ since the span of the previous rows has at most $t - 1$ dimensions, but the convex hull of $\boldsymbol{e}_{[n]}^{(t)}$ has at least $n$ linearly independent elements ($r \leq n$), so it is not covered by the previous $t - 1$ rows. Therefore, there exists $\{\boldsymbol{\theta}_1^{(t)}\}_{t=1}^r$ such that the resulting $\boldsymbol{Z}$ matrix has rank at least $r$. Then, using the result of Claim 1, there exists Attention weights $\boldsymbol{W}_{K1}, \boldsymbol{W}_{Q1}$ that produce the desired set of convex combinations $\{\boldsymbol{\theta}_1^{(t)}\}_{t=1}^r$, and subsequently the first $r$ rows of the desired feature matrix $\boldsymbol{Z}$.

*Step 2. Induction Step.* Next, assume that for $H = H'$, there exists heads with Attention weights $\{(\boldsymbol{W}_{Kh}, \boldsymbol{W}_{Qh})\}_{h=1}^{H'}$ such that matrix $\boldsymbol{Z}$ has rank $H'(r - 1) + 1$. We prove for $H = H' + 1$ there exists Attention weights $\{(\boldsymbol{W}_{Kh}, \boldsymbol{W}_{Qh})\}_{h=1}^{H'+1}$ such that the resulting matrix $\boldsymbol{Z}$ attains rank $(H' + 1)(r - 1) + 1$. Let us refer to the matrices derived from the induction hypothesis by a prime superscript (*i.e.*, $\boldsymbol{Z}'$, and $\{(\boldsymbol{W}_{Kh}', \boldsymbol{W}_{Qh}')\}_{h=1}^{H'}$). Now, for the case of $H = H' + 1$, we set the weights of the first $H'$ Attention heads to be equal to that of $H'$. Namely,

$$\boldsymbol{W}_{Kh} := \boldsymbol{W}_{Kh}', \quad \boldsymbol{W}_{Qh} := \boldsymbol{W}_{Qh}'$$

for all $h \in [H']$. Then, the feature matrix $\boldsymbol{Z}$ can be written as the following block form:

$$\boldsymbol{Z} = \left[\ \underbrace{\boldsymbol{Z}'}_{H'd}\ \middle|\ \underbrace{\boldsymbol{Z}''}_{d}\ \right],$$

where $\boldsymbol{Z}' \in \mathbb{R}^{T \times dH'}$ is the matrix derived from the hypothesis and $\boldsymbol{Z}'' \in \mathbb{R}^{T \times d}$ is the block corresponding to the new head $H' + 1$, whose parameters we have not yet determined. Now, we know $\text{rank}(\boldsymbol{Z}') \geq H'(r - 1) + 1$. Define $k := \text{rank}(\boldsymbol{Z}')$. WLOG, assume that the first $k$ rows of the matrix $\boldsymbol{Z}'$ are linearly independent (otherwise, we can rearrange the rows of $\boldsymbol{Z}$, *i.e.* the examples, to satisfy this property). Now, take the last $r - 1$ rows that are linearly dependent on the first $k$ rows. We defer the case for when less than $r - 1$ such rows exist (*i.e.*, $k + r - 1 > T$) to the later stages of the proof.

Let us partition the matrix $\boldsymbol{Z}$ into two blocks: the first $T - r + 1$ rows and the last $r - 1$ rows. Namely,

$$\boldsymbol{Z} = \left[ \frac{\boldsymbol{Z}_a}{\boldsymbol{Z}_b} \right] = \left[ \begin{array}{c|c} \boldsymbol{Z}'_a & \boldsymbol{Z}''_a \\ \hline \boldsymbol{Z}'_b & \boldsymbol{Z}''_b \end{array} \right]. \tag{9}$$

Observe that the rows of $\boldsymbol{Z}'_b$ are all linear combinations of rows of $\boldsymbol{Z}'_a$. So, we can zero out the $\boldsymbol{Z}'_b$ block by adding a linear combination of rows of $\boldsymbol{Z}'_a$. Such operations add a combination of rows of $\boldsymbol{Z}''_a$ to the $\boldsymbol{Z}''_b$ as well. Notice that the rank of $\boldsymbol{Z}$ stays the same since such operations do not change rank. So let us call $\boldsymbol{Z}_{op}$ the resulting matrix:

$$\boldsymbol{Z}_{op} = \left[ \begin{array}{c|c} \boldsymbol{Z}'_a & \boldsymbol{Z}''_a \\ \hline \boldsymbol{0} & \boldsymbol{Z}''_b + \boldsymbol{A} \end{array} \right],$$

where

$$\boldsymbol{A} := \boldsymbol{A}(\boldsymbol{Z}'_a, \boldsymbol{Z}'_b, \boldsymbol{Z}''_a) := \boldsymbol{C}(\boldsymbol{Z}'_a, \boldsymbol{Z}'_b)\boldsymbol{Z}''_a \in \mathbb{R}^{(r-1) \times d}$$

contains a linear combination of rows of $\boldsymbol{Z}''_a$ (whose coefficients are in $\boldsymbol{C}(\boldsymbol{Z}'_a, \boldsymbol{Z}'_b)$ and are a function of $\boldsymbol{Z}'_a, \boldsymbol{Z}'_b$). Since

$$\operatorname{rank}(\boldsymbol{Z}) = \operatorname{rank}(\boldsymbol{Z}_{op}) \overset{\text{Lemma 1}}{\geq} \operatorname{rank}(\boldsymbol{Z}''_b + \boldsymbol{A}) + \operatorname{rank}(\boldsymbol{Z}'_a) \geq \operatorname{rank}(\boldsymbol{Z}''_b + \boldsymbol{A}) + H'(r-1) + 1,$$

we have to find suitable Attention weights such that $\operatorname{rank}(\boldsymbol{Z}''_b + \boldsymbol{A}) \geq r - 1$ to achieve a $(H' + 1)(r-1) + 1$ rank for $\boldsymbol{Z}$ and complete the proof.

Now, consider Claim 2 for some fixed convex combination vectors $\{\boldsymbol{\theta}_+^{(t)}\}_{t \in \mathcal{R}^c}$, and any desired $\{\boldsymbol{\theta}_{H'+1}^{(t)}\}_{t \in \mathcal{R}}$. For any $\delta > 0$, the claim statement gives key/query weights, and subsequently a resulting intermediate matrix $\boldsymbol{Z}_\delta$. Let us define $\boldsymbol{Z}_+$ as the *limiting matrix* $\lim_{\delta \to 0^+} \boldsymbol{Z}_\delta$, and similarly define its sub-blocks corresponding to this head by the same sign, *i.e.*, $\boldsymbol{Z}''_{a,+}, \boldsymbol{Z}''_{b,+}, \boldsymbol{A}_+$, and similarly $\boldsymbol{Z}''_{a,\delta}, \boldsymbol{Z}''_{b,\delta}, \boldsymbol{A}_\delta$. Notice that $\boldsymbol{A}_+$ is only a function of $\boldsymbol{Z}'_a, \boldsymbol{Z}'_b, \boldsymbol{Z}''_{a,+}$. As a result, different choices of $\{\boldsymbol{\theta}_{H'+1}^{(t)}\}_{t \in \mathcal{R}}$ produce the same $\boldsymbol{A}_+$ since $\{\boldsymbol{\theta}_+^{(t)}\}_{t \in \mathcal{R}^c}$ remains fixed under Claim 2. Therefore, we first find $\boldsymbol{\theta}_{H'+1}^{(t)}$ such that $\operatorname{rank}(\boldsymbol{Z}''_{b,+} + \boldsymbol{A}_+) \geq r - 1$, then we find small enough $\delta$ that properly approximates the limiting matrices. Similar to the base of induction (Step 1), we inductively build the individual rows of $\boldsymbol{Z}''_{b,+}$, but the difference is that the original matrix $\boldsymbol{Z}''_{b,+}$ is added to a fixed matrix $\boldsymbol{A}_+$. Hence, at each inductive step $t$, the following modified constraint must be satisfied:

$$\left[\boldsymbol{Z}''_{b,+} + \boldsymbol{A}_+\right]_t \notin \operatorname{span}\left\{\left[\boldsymbol{Z}''_{b,+} + \boldsymbol{A}_+\right]_1, \left[\boldsymbol{Z}''_{b,+} + \boldsymbol{A}_+\right]_2, \ldots, \left[\boldsymbol{Z}''_{b,+} + \boldsymbol{A}_+\right]_{t-1}\right\},$$

which can be satisfied by a stronger constraint:

$$\left[\boldsymbol{Z}''_{b,+}\right]_t \notin \operatorname{span}\left\{\left[\boldsymbol{A}_+\right]_t, \left[\boldsymbol{Z}''_{b,+} + \boldsymbol{A}_+\right]_1, \left[\boldsymbol{Z}''_{b,+} + \boldsymbol{A}_+\right]_2, \ldots, \left[\boldsymbol{Z}''_{b,+} + \boldsymbol{A}_+\right]_{t-1}\right\}.$$

Hence, since the input context tokens are in an $n$ dimensional space, the set of feasible valid rows $[\boldsymbol{Z}''_{b,+}]_t$ lies in an $n$ dimensional space as well, which allows us to continue for $n - 1, (n \geq r)$ rows (and not $n$, due to presence of $[\boldsymbol{A}_+]_t$), find suitable $\{\boldsymbol{\theta}_{H'+1}^{(t)}\}_{t \in \mathcal{R}}$, and build a full-rank matrix $\boldsymbol{Z}''_{b,+}$ with $\operatorname{rank}(\boldsymbol{Z}''_{b,+} + \boldsymbol{A}_+) \geq r - 1$. Notice that no well-defined key/weights generate $\boldsymbol{Z}_+$ as the weights need to be infinite (see proof of Claim 2). However, we only need a small enough $\delta$ such that

$$\operatorname{rank}(\boldsymbol{Z}''_{b,\delta} + \boldsymbol{A}_\delta) \geq \operatorname{rank}(\boldsymbol{Z}''_{b,+} + \boldsymbol{A}_+) \geq r - 1 \tag{10}$$

for the weights to be well-defined. In Appendix B.2 we prove the existence of such $\delta$. Therefore, there exists $\boldsymbol{W}_{K\,H'+1}, \boldsymbol{W}_{Q\,H'+1}$ that produces $\operatorname{rank}(\boldsymbol{Z}''_b + \boldsymbol{A}) \geq r - 1$, and the proof is complete.

*The case of less than $r - 1$ linearly dependent rows of $\boldsymbol{Z}'$.* In this case, we have $k + r - 1 > T$. Let us define $m := T - k < r - 1$. Since we have assumed $T \geq (H' + 1)(r-1) + 1$, this means that that $\boldsymbol{Z}'$ already has a rank at least $(H' + 1)(r-1) + 1 - m$. So, the head index $H' + 1$ needs to increase the rank of $\boldsymbol{Z}'$ by $m$. Since $m \leq r - 1$, all the above steps also apply to $m$ and the rest of the proof procedure is the same.

### B.1 PROOF OF CLAIMS 1 AND 2

First, note each $\boldsymbol{\theta}_{h,*}^{(t)}$ is derived from applying Softmax nonlinearity on the logits $\boldsymbol{\alpha}_{h,*}^{(t)}$. So, we need to solve a system of equations for $\boldsymbol{\alpha}_{h,*}^{(t)}$ for each $t \in [T]$[4]. Since our focus is on a *single* head, we omit the subscript $h$ for simplicity. Furthermore, we define $\boldsymbol{W} := \boldsymbol{W}_K \boldsymbol{W}_Q^\top$. Now, for data indices $t_1 < t_2 < \cdots < t_m \in [T]$ where $m \leq n$, we are interested in solving the following system of equations:

$$\boldsymbol{E}^{(t_i)} \boldsymbol{W} \boldsymbol{e}^{(t_i)} = \boldsymbol{\alpha}_*^{(t_i)} \quad \forall i \in [m].$$

Since each $\boldsymbol{E}^{(t_i)} \in \mathbb{R}^{n \times d}$ is full-rank with rank $n$, it has a pseudo right inverse $\boldsymbol{E}^{(t_i)^\dagger} \in \mathbb{R}^{d \times n}$ such that $\boldsymbol{E}^{(t_i)} \boldsymbol{E}^{(t_i)^\dagger} = \boldsymbol{I}$ . So, we solve the following system of equations

$$\boldsymbol{W} \boldsymbol{e}^{(t_i)} = \boldsymbol{E}^{(t_i)^\dagger} \boldsymbol{\alpha}_*^{(t_i)} \quad \forall i \in [m],$$

$$\Rightarrow \boldsymbol{E}^{(t_i)} \boldsymbol{W} \boldsymbol{e}^{(t_i)} = \boldsymbol{\alpha}_*^{(t_i)} \quad \forall i \in [m].$$

Now, let us define $\boldsymbol{\beta}_*^{(t_i)} := \boldsymbol{E}^{(t_i)^\dagger} \boldsymbol{\alpha}_*^{(t_i)}$. Then, we get the following linear system of equations to solve:

$$\boldsymbol{W} \begin{bmatrix} \boldsymbol{e}^{(t_1)}, & \boldsymbol{e}^{(t_2)}, & \ldots, & \boldsymbol{e}^{(t_m)} \end{bmatrix} = \begin{bmatrix} \boldsymbol{\beta}_*^{(t_1)}, & \boldsymbol{\beta}_*^{(t_2)}, & \ldots, & \boldsymbol{\beta}_*^{(t_m)} \end{bmatrix}.$$

Due to Assumption 1 the query tokens have Kruskal Rank at least $n$, and $m \leq r \leq n$, so the query token matrix in the left-hand-side above has a pseudo left inverse. Thus, $\boldsymbol{W}$ has at least one solution with rank at most $m$. Assuming the solution is $\boldsymbol{W}^* \in \mathbb{R}^{d \times d}$, there exists $\boldsymbol{W}_{Kh}, \boldsymbol{W}_{Qh} \in \mathbb{R}^{d \times d_h}$ such that $\boldsymbol{W}_{Kh} \boldsymbol{W}_{Qh}^\top = \boldsymbol{W}^*$ for the weights of the Attention head. So, we can prove Claim 1 by setting $m = r$ and $(t_1, t_2, \ldots t_r) = (1, 2, \ldots, r)$.

Next, to prove Claim 2, we prove that if $m < r \leq n$, $(t_1, t_2, \ldots, t_m) = (T-m+1, T-m+2, \ldots, T)$, then we can not only achieve any desired logits for the examples $(T - m + 1, T - m + 2, \ldots, T)$, but also keep the $\{\boldsymbol{\theta}^{(t)}\}_{t=1}^{T-m}$ approximately unchanged. So far, we have found a $\boldsymbol{W}^*$ that achieves the set of desired logits. Now, there exists a rank-one linear transformation $\boldsymbol{W}^+ \in \mathbb{R}^{d \times d}$ such that

$$\boldsymbol{E}^{(t)} \boldsymbol{W}^+ \boldsymbol{e}^{(t)} = \boldsymbol{0} \quad \text{for all } T - m < t \leq T, \tag{11}$$

$$\left[ \boldsymbol{E}^{(t)} \boldsymbol{W}^+ \boldsymbol{e}^{(t)} \right]_i \neq \left[ \boldsymbol{E}^{(t)} \boldsymbol{W}^+ \boldsymbol{e}^{(t)} \right]_j \quad \text{for all } 1 \leq t \leq T - m, \ i, j \in [n], i \neq j. \tag{12}$$

This is due to Lemma 3.

So, we set $\boldsymbol{W} := \boldsymbol{W}^* + c\boldsymbol{W}^+$, and choose a large enough $c$. This way, the desired logits for $T - m < t \leq T$ stay the same since $c\boldsymbol{W}^+$ has no effect on them, while for the rest of logits, $c\boldsymbol{W}^+$ dominates $\boldsymbol{W}^*$ and saturates the Softmax operator. Lemma 4 proves the existence of such $c$, so, we can prove Claim 2 by finding $\boldsymbol{W}_{Kh}, \boldsymbol{W}_{Qh} \in \mathbb{R}^{d \times d_h}$ such that $\boldsymbol{W}_{Kh} \boldsymbol{W}_{Qh}^\top = \boldsymbol{W}^* + c\boldsymbol{W}^+$. Note that this construction is feasible because $\text{rank}(\boldsymbol{W}^* + c\boldsymbol{W}^+) \leq m + 1 \leq r \leq d_h$.

### B.2 PROOF OF EXISTENCE OF $\delta$ FOR EQ. (10)

Let us call the $\boldsymbol{\theta}_h^{(t)}$ derived from Claim 2 as $\boldsymbol{\theta}_\delta^{(t)}$. So, for any $\delta > 0$ we have:

$$\left\| \boldsymbol{\theta}_\delta^{(t)} - \boldsymbol{\theta}_+^{(t)} \right\|_1 < \delta, \text{ for all } t \in \mathcal{R}^c$$

$$\boldsymbol{\theta}_\delta^{(t)} = \boldsymbol{\theta}_+^{(t)} = \boldsymbol{\theta}_*^{(t)}, \text{ for all } t \in \mathcal{R}$$

$$\Rightarrow \boldsymbol{Z}_{b,\delta}'' = \boldsymbol{Z}_{b,+}''.$$

To prove the rank inequality argument, we use a corollary of the Weyl's singular value theorem (Lemma 5) and prove:

$$\Delta_\delta := \left( \boldsymbol{Z}_{b,\delta}'' + \boldsymbol{A}_\delta \right) - \left( \boldsymbol{Z}_{b,+}'' + \boldsymbol{A}_+ \right), \quad \sigma_{\max}(\Delta_\delta) < \sigma_{\min}(\boldsymbol{Z}_{b,+}'' + \boldsymbol{A}_+), \tag{13}$$

which in turn leads to $\text{rank}(\boldsymbol{Z}_{b,\delta}'' + \boldsymbol{A}_\delta) \geq \text{rank}(\boldsymbol{Z}_{b,+}'' + \boldsymbol{A}_+)$. First, let us assume $\boldsymbol{Z}_{b,+}'' + \boldsymbol{A}_+$ is not a zero matrix (otherwise we must have $r - 1 = 0$, in which case any $\delta > 0$ would satisfy the

---

[4] The set of logits that produce a convex combination is not necessarily unique. However, uniqueness is not a requirement in this case, as we only require the existence of some set of logits that produces a convex combination.

inequality). We now bound the spectral norm of the perturbation matrix $\Delta_\delta$, *i.e.*, $\sigma_{\max}(\Delta_\delta)$.

$$
\begin{aligned}
\sigma_{\max}(\Delta_\delta) &= \|\Delta_\delta\|_2 \\
&= \|\boldsymbol{A}_\delta - \boldsymbol{A}_+\|_2 \qquad\qquad (\boldsymbol{Z}''_{b,+} = \boldsymbol{Z}''_{b,\delta}) \\
&= \left\|\boldsymbol{C}(\boldsymbol{Z}'_a, \boldsymbol{Z}'_b)\boldsymbol{Z}''_{a,\delta} - \boldsymbol{C}(\boldsymbol{Z}'_a, \boldsymbol{Z}'_b)\boldsymbol{Z}''_{a,+}\right\|_2 \\
&= \left\|\boldsymbol{C}(\boldsymbol{Z}'_a, \boldsymbol{Z}'_b)\left(\boldsymbol{Z}''_{a,\delta} - \boldsymbol{Z}''_{a,+}\right)\right\|_2 \\
&\leq \left\|\boldsymbol{C}(\boldsymbol{Z}'_a, \boldsymbol{Z}'_b)\right\|_2 \left\|\boldsymbol{Z}''_{a,\delta} - \boldsymbol{Z}''_{a,+}\right\|_2 \\
&\leq \left\|\boldsymbol{C}(\boldsymbol{Z}'_a, \boldsymbol{Z}'_b)\right\|_2 \left\|\boldsymbol{Z}''_{a,\delta} - \boldsymbol{Z}''_{a,+}\right\|_F,
\end{aligned}
$$

where $\boldsymbol{C}(\boldsymbol{Z}'_a, \boldsymbol{Z}'_b)$ is a constant matrix containing coefficients derived from zeroing out $\boldsymbol{Z}'_b$ using the rows of $\boldsymbol{Z}'_a$. To bound the Frobenius norm of $\boldsymbol{Z}''_{a,\delta} - \boldsymbol{Z}''_{a,+}$, notice that $t$-th row of this matrix is:

$$
\begin{aligned}
\left\|\left[\boldsymbol{Z}''_{a,\delta} - \boldsymbol{Z}''_{a,+}\right]_t\right\|_2 &= \left\|\boldsymbol{E}^{(t)\top}\boldsymbol{\theta}_\delta^{(t+T')} - \boldsymbol{E}^{(t)\top}\boldsymbol{\theta}_+^{(t)}\right\|_2 \\
&= \left\|\boldsymbol{E}^{(t)\top}\left(\boldsymbol{\theta}_\delta^{(t)} - \boldsymbol{\theta}_+^{(t)}\right)\right\|_2 \\
&\leq \left\|\boldsymbol{E}^{(t)\top}\right\|_{1,2}\left\|\left(\boldsymbol{\theta}_\delta^{(t)} - \boldsymbol{\theta}_+^{(t)}\right)\right\|_1 \\
&\leq e_{\max}\delta,
\end{aligned}
$$

where $e_{\max}$ is the maximum token Euclidian norm of all tokens in the dataset. Therefore:

$$
\left\|\boldsymbol{Z}''_{a,\delta} - \boldsymbol{Z}''_{a,+}\right\|_F \leq \sqrt{T'}e_{\max}\delta.
$$

Hence, choosing

$$
\delta \leq \frac{\sigma_{\min}(\boldsymbol{Z}''_{b,+} + \boldsymbol{A}_+)}{\sqrt{T'}e_{\max}\left\|\boldsymbol{C}(\boldsymbol{Z}'_a, \boldsymbol{Z}'_b)\right\|_2}
$$

gives the desired inequality in Eq. (13).

## C EXTENSION OF THEOREM 1 TO ATTENTION WITH SKIP CONNECTION

Theorem 1 holds true for when we have Attention with skip connection. Recall that skip connection keeps all the computational mechanisms of MHA the same except for Eq. (5), where the updated equation is:

$$
\boldsymbol{o} := \boldsymbol{W}_O^\top[\boldsymbol{p}_1; \boldsymbol{p}_2; \ldots; \boldsymbol{p}_H] + \boldsymbol{e} \qquad\qquad (\boldsymbol{o} \in \mathbb{R}^d)
$$

The only modification that we have to make, is to modify the labels. Let us consider the setting of Theorem 1, with dataset $\mathcal{T} = \left\{\left(\boldsymbol{E}^{(t)}, \boldsymbol{e}^{(t)}, \boldsymbol{y}^{(t)}\right)\right\}_{t=1}^T$ and Attention weights $\mathcal{W}$. Now, if we add the skip connection to the Attention layer with the same set of weights, the new $\hat{\boldsymbol{y}}^{(t)}$ will be:

$$
\hat{\boldsymbol{y}}^{(t)} = \boldsymbol{W}_D^\top(\boldsymbol{e}^{(t)} + \boldsymbol{o}^{(t)}) = \boldsymbol{I}_{d_{\text{out}} \times d}(\boldsymbol{e}^{(t)} + \boldsymbol{o}^{(t)}) = \boldsymbol{y}^{(t)} + \boldsymbol{I}_{d_{\text{out}} \times d}\boldsymbol{e}^{(t)}.
$$

Therefore, we get labels $\boldsymbol{y}^{(t)} + \boldsymbol{I}_{d_{\text{out}} \times d}\boldsymbol{e}^{(t)}$, which is the true label shifted by the first $d_{\text{out}}$ dimensions of the query vector. So, we need to define a new dataset

$$
\mathcal{T}' = \left\{\left(\boldsymbol{E}^{(t)}, \boldsymbol{e}^{(t)}, \boldsymbol{y}^{(t)} - \boldsymbol{I}_{d_{\text{out}} \times d}\boldsymbol{e}^{(t)}\right)\right\}_{t=1}^T,
$$

and find Attention weights $\mathcal{W}'$ for $\mathcal{T}'$ according to Theorem 1. Then, we can use the set of Attention weights $\mathcal{W}'$ to get the correct label $\boldsymbol{y}^{(t)} + \boldsymbol{I}_{d_{\text{out}} \times d}\boldsymbol{e}^{(t)} - \boldsymbol{I}_{d_{\text{out}} \times d}\boldsymbol{e}^{(t)}$ for an Attention layer with skip connection and dataset $\mathcal{T}$.

## D ADDITIONAL LEMMAS

**Lemma 1.** *For any real-valued block matrix $\boldsymbol{X}$ of the form*

$$
\boldsymbol{X} = \left[\begin{array}{cc} \boldsymbol{A} & \boldsymbol{C} \\ \boldsymbol{0} & \boldsymbol{B} \end{array}\right]
$$

*we have* $\operatorname{rank}(\boldsymbol{X}) \geq \operatorname{rank}(\boldsymbol{A}) + \operatorname{rank}(\boldsymbol{B})$.

**Lemma 2.** *Assume vectors $\boldsymbol{e}_1, \boldsymbol{e}_2, \ldots, \boldsymbol{e}_n \in \mathbb{R}^d$ are linearly independent. Then,*

$$\boldsymbol{e}_i' := \boldsymbol{e}_i + \frac{1}{n}(\boldsymbol{e}_1 + \boldsymbol{e}_2 + \cdots + \boldsymbol{e}_n),$$

*are also linearly independent for all $i \in [n]$.*

*Proof.* Due to the linear independent assumption, the only solution to

$$a_1 \boldsymbol{e}_1 + a_2 \boldsymbol{e}_2 + \cdots + a_n \boldsymbol{e}_n = \boldsymbol{0} \quad s.t. \quad a_1, a_2, \ldots, a_n \in \mathbb{R}$$

is $a_1 = a_2 = \cdots = a_n = 0$. Now, let us examine the same system of equations for $\boldsymbol{e}_i'$. Namely,

$$b_1 \boldsymbol{e}_1' + b_2 \boldsymbol{e}_2' + \cdots + b_n \boldsymbol{e}_n' = \boldsymbol{0} \quad s.t. \quad b_1, b_2, \ldots, b_n \in \mathbb{R}.$$

We prove zero is the only solution to this system to conclude the proof.

$$b_1 \boldsymbol{e}_1' + b_2 \boldsymbol{e}_2' + \cdots + b_n \boldsymbol{e}_n'$$

$$= b_1 \boldsymbol{e}_1 + b_2 \boldsymbol{e}_2 + \cdots + b_n \boldsymbol{e}_n + \underbrace{\frac{1}{n}(b_1 + b_2 + \cdots + b_n)}_{\bar{b}}(\boldsymbol{e}_1 + \boldsymbol{e}_2 + \cdots + \boldsymbol{e}_n)$$

$$= (b_1 + \bar{b})\boldsymbol{e}_1 + (b_2 + \bar{b})\boldsymbol{e}_2 + \cdots + (b_n + \bar{b})\boldsymbol{e}_n = \boldsymbol{0}.$$

According to the linear independence assumption, we must have

$$b_i + \bar{b} = 0 \quad \text{for all } i \in [n].$$

By summing all the terms up, we conclude that $b_i = 0$ for all $i \in [n]$ is the only solution and this concludes the proof. $\square$

**Lemma 3.** *Consider a set of full-rank matrices $\boldsymbol{E}^{(1)}, \boldsymbol{E}^{(2)}, \ldots, \boldsymbol{E}^{(T)} \in \mathbb{R}^{n \times d}$ where $n < d$. Moreover, consider a set of vectors $\boldsymbol{e}^{(1)}, \boldsymbol{e}^{(2)}, \ldots, \boldsymbol{e}^{(T)} \in \mathbb{R}^d$ with a Kruskal Rank at least $n$. Then, for any $m \leq n - 1$ there exists a rank-one matrix $\boldsymbol{W} \in \mathbb{R}^{d \times d}$ such that*

$$\boldsymbol{E}^{(t)} \boldsymbol{W} \boldsymbol{e}^{(t)} = \boldsymbol{0} \quad \text{for all } 1 \leq t \leq m, \tag{14}$$

$$\left[\boldsymbol{E}^{(t)} \boldsymbol{W} \boldsymbol{e}^{(t)}\right]_i \neq \left[\boldsymbol{E}^{(t)} \boldsymbol{W} \boldsymbol{e}^{(t)}\right]_j \quad \text{for all } m < t \leq T, \, i, j \in [n], i \neq j. \tag{15}$$

*Proof.* Let us define a basis for $\mathbb{R}^d$ as $\boldsymbol{b}_1, \boldsymbol{b}_2, \boldsymbol{b}_3, \ldots, \boldsymbol{b}_d \in \mathbb{R}^d$ such that $\boldsymbol{b}_i = \boldsymbol{e}^{(i)}$ for all $i \in [m]$, and $\boldsymbol{b}_{m+1}, \ldots \boldsymbol{b}_{m+2}, \ldots, \boldsymbol{b}_d$ are an arbitrary extension of $\boldsymbol{e}_{[m]}$ to a basis. Then, a linear transformation $\boldsymbol{W}$ is uniquely determined by defining $\boldsymbol{c}_1, \boldsymbol{c}_2, \ldots \boldsymbol{c}_d \in \mathbb{R}^d$ and setting

$$\boldsymbol{W} \boldsymbol{b}_i = \boldsymbol{c}_i \quad \text{for all } i \in [d].$$

First, to satisfy Eq. (14), we set $\boldsymbol{c}_i := \boldsymbol{0}$ for all $i \in [m]$, which gives $\boldsymbol{W} \boldsymbol{b}_i = \boldsymbol{W} \boldsymbol{e}^{(i)} = \boldsymbol{c}_i = \boldsymbol{0}$ for all $i \in [m]$. Let us decompose every vector $\boldsymbol{e}^{(t)}$ into the mentioned basis, *i.e.*,

$$\boldsymbol{e}^{(t)} = a_1^{(t)} \boldsymbol{b}_1 + a_2^{(t)} \boldsymbol{b}_2 + \cdots + a_d^{(t)} \boldsymbol{b}_d \quad \text{for all } m + 1 \leq t \leq T.$$

Since $\{\boldsymbol{e}^{(t)}\}_{t=1}^T$ have a Kruskal Rank at least $n$, then for any $t \in [T]$ at least one of $a_{m+1}^{(t)}, a_{m+2}^{(t)}, \ldots, a_d^{(t)}$ must be nonzero, otherwise, $\boldsymbol{e}^{(t)}$ can be written as a linear combination of the first $m$ vectors. Moreover, there exists fixed $u_{m+1}, u_{m+2}, \ldots, u_d \in \mathbb{R}$ such that

$$a_{m+1}^{(t)} u_{m+1} + a_{m+2}^{(t)} u_{m+2} \cdots + a_d^{(t)} u_d \neq 0 \quad \text{for all } m + 1 \leq t \leq d,$$

for instance, we can choose $u_i$ *iid* from a Standard Gaussian Distribution and satisfy this property almost surely. Now, let us define

$$\boldsymbol{c} \sim \mathcal{N}(\boldsymbol{0}, \boldsymbol{I}) \tag{16}$$

$$\boldsymbol{c}_i := u_i \boldsymbol{c} \quad \text{for all } m + 1 \leq i \leq d. \tag{17}$$

Therefore,

$$\boldsymbol{W} \boldsymbol{e}^{(t)} = a_1^{(t)} \boldsymbol{W} \boldsymbol{b}_1 + a_2^{(t)} \boldsymbol{W} \boldsymbol{b}_2 + \cdots + a_d^{(t)} \boldsymbol{W} \boldsymbol{b}_d \tag{18}$$

$$= a_1^{(t)} \boldsymbol{c}_1 + a_2^{(t)} \boldsymbol{c}_2 + \cdots + a_d^{(t)} \boldsymbol{c}_d \tag{19}$$

$$= a_{m+1}^{(t)} \boldsymbol{c}_{m+1} + a_{m+2}^{(t)} \boldsymbol{c}_{m+2} \cdots + a_d^{(t)} \boldsymbol{c}_d \tag{20}$$

$$= \left(a_{m+1}^{(t)} u_{m+1} + a_{m+2}^{(t)} u_{m+2} \cdots + a_d^{(t)} u_d\right) \boldsymbol{c} \quad \text{for all } m + 1 \leq t \leq T, \tag{21}$$

with one nonzero coefficient. Now, since $\boldsymbol{c}$ is standard gaussian, then

$$\boldsymbol{W}\boldsymbol{e}^{(t)} \sim \mathcal{N}\left(\boldsymbol{0}, \left(a_{m+1}^{(t)}u_{m+1} + a_{m+2}^{(t)}u_{m+2}\cdots + a_d^{(t)}u_d\right)^2 \boldsymbol{I}\right).$$

Moreover, $\boldsymbol{E}^{(t)}$ is full-rank, so none of its rows are equal to each other. As a result,

$$\left(\boldsymbol{e}_i^{(t)} - \boldsymbol{e}_j^{(t)}\right)^\top \boldsymbol{W}\boldsymbol{e}^{(t)}$$

follows a Gaussian distribution for all $i, j \in [n]$, $i \neq j$. Since $\{0\}$ has zero Lebesgue measure, the probability of having two equal elements is zero. So, Eq. 15 is satisfied almost surely. Note that this construction produces a rank-one $\boldsymbol{W}$ since the rank of $[\boldsymbol{c}_1, \boldsymbol{c}_2, \dots, \boldsymbol{c}_d]$ is at most one. Therefore, there exists a rank-one matrix $\boldsymbol{W}$ that satisfies Eq. (15) and this completes the proof.

$\square$

**Lemma 4.** *Consider an input matrix $\boldsymbol{E} \in \mathbb{R}^{n \times d}$, an input vector $\boldsymbol{e} \in \mathbb{R}^d$, and a weight matrix $\boldsymbol{W}^+ \in \mathbb{R}^{d \times d}$ such that*

$$\left[\boldsymbol{E}\boldsymbol{W}^+\boldsymbol{e}\right]_i \neq \left[\boldsymbol{E}\boldsymbol{W}^+\boldsymbol{e}\right]_j \quad \text{for all } i, j \in [n], i \neq j. \tag{22}$$

*For any $\boldsymbol{W}^* \in \mathbb{R}^{d \times d}$ define $\boldsymbol{W}_c := \boldsymbol{W}^* + c\boldsymbol{W}^+$ where $c \in \mathbb{R}$ is a scalar. Then,*

$$\boldsymbol{\theta}_+ := \lim_{c \to \infty} \boldsymbol{\theta}_c := \lim_{c \to \infty} \text{Softmax}\left([\boldsymbol{E}\boldsymbol{W}_c\boldsymbol{e}]\right),$$

*is only a function of $\boldsymbol{W}^+$. Consequently, $\boldsymbol{\theta}_c$ converges to $\boldsymbol{\theta}_+$ in $L^1$ sense, i.e., for any $\delta > 0$, there exists $c_\delta$ such that*

$$\|\boldsymbol{\theta}_+ - \boldsymbol{\theta}_{c_\delta}\|_1 < \delta.$$

*Proof.* Let us define

$$\boldsymbol{\alpha}^+ := \boldsymbol{E}\boldsymbol{W}^+\boldsymbol{e}, \quad \text{and} \quad \boldsymbol{\alpha}^* := \boldsymbol{E}\boldsymbol{W}^*\boldsymbol{e}.$$

Now, $i$th element of $\boldsymbol{\theta}$ is given by:

$$
\begin{aligned}
[\boldsymbol{\theta}_+]_i &= \lim_{c \to \infty} [\boldsymbol{\theta}_c]_i = \lim_{c \to \infty} \frac{\exp\left(\boldsymbol{\alpha}_i^* + c\boldsymbol{\alpha}_i^+\right)}{\sum_{j \in [n]} \exp\left(\boldsymbol{\alpha}_j^* + c\boldsymbol{\alpha}_j^+\right)} \\
&= \lim_{c \to \infty} \frac{\exp\left(\boldsymbol{\alpha}_i^*\right)}{\exp\left(\boldsymbol{\alpha}_i^*\right) + \sum_{j \neq i} \exp\left(\boldsymbol{\alpha}_j^* + c(\boldsymbol{\alpha}_j^+ - \boldsymbol{\alpha}_i^+)\right)} \\
&= \mathbf{1}\left[i = \arg\max\{\boldsymbol{\alpha}_1^+, \boldsymbol{\alpha}_2^+, \dots, \boldsymbol{\alpha}_n^+\}\right],
\end{aligned}
$$

where the last equality is due to $\boldsymbol{\alpha}_i^+$ being distinct. Therefore, $[\boldsymbol{\theta}_+]_i$ is a one-hot vector determined by $\boldsymbol{W}^+$. Consequently, since each individual entry of $\boldsymbol{\theta}_c$ converges to $\boldsymbol{\theta}_+$, we get convergence in $L_1$ and this completes the proof. $\square$

**Lemma 5.** *Let $\boldsymbol{A}, \boldsymbol{B} \in \mathbb{R}^{n \times m}$ be two matrices satisfying*

$$\sigma_{\max}(\boldsymbol{B} - \boldsymbol{A}) < \sigma_{\min}(\boldsymbol{A}),$$

*where $\sigma_{\min}, \sigma_{\max}$ are the smallest and largest non-zero singular values. Then, we have*

$$\text{rank}(\boldsymbol{B}) \geq \text{rank}(\boldsymbol{A}).$$

*Proof.* From Weyl's singular value theorem (Horn & Johnson, 1990), we know

$$|\sigma_k(\boldsymbol{A} + \Delta) - \sigma_k(\boldsymbol{A})| \leq \sigma_{\max}(\Delta),$$

for any perturbation matrix $\Delta$, and singular value index $k \in [\min(n, m)]$. Setting $\Delta = \boldsymbol{B} - \boldsymbol{A}$ and using the lemma's assumption we have:

$$|\sigma_k(\boldsymbol{B}) - \sigma_k(\boldsymbol{A})| \leq \sigma_{\max}(\boldsymbol{B} - \boldsymbol{A}) < \sigma_{\min}(\boldsymbol{A}) \leq \sigma_k(\boldsymbol{A})$$

for all $k$ such that $\sigma_k(\boldsymbol{A}) > 0$. Hence, for any non-zero singular value $k$, $\boldsymbol{B}$ has a corresponding non-zero singular value and $\text{rank}(\boldsymbol{B}) \geq \text{rank}(\boldsymbol{A})$. $\square$

## E  PROOF OF PROPOSITION 2

First, note that when the contexts are shared, the matrix $\boldsymbol{Z}$ has the following decomposition:

$$\boldsymbol{Z}^\top = \left(\boldsymbol{I}_H \otimes \boldsymbol{E}^T\right) \boldsymbol{\Theta}^\top,$$

where $\otimes$ is the Kronecker product and $\boldsymbol{\Theta} \in \mathbb{R}^{T \times Hn}$ is defined similar to $\boldsymbol{Z}$ (*i.e.*, the block in $t$-th row and $h$-th column is $\boldsymbol{\theta}_h^{(t)^\top}$). Notice that $\boldsymbol{\Theta} \in \mathbb{R}^{T \times Hn}$, therefore, the rank of $\boldsymbol{Z}$ can be at most $Hn$. However, we prove here that the column rank of $\boldsymbol{\Theta}$ is at most $H(n-1)+1$ due to the structure of $\boldsymbol{\theta}_h^{(t)}$ being convex combination vectors. With a slight abuse of notation, let us write $\boldsymbol{\Theta}$ as:

$$\boldsymbol{\Theta} = [\boldsymbol{\Theta}_1, \quad \boldsymbol{\Theta}_2, \quad \boldsymbol{\Theta}_3 \quad \dots, \quad \boldsymbol{\Theta}_H],$$

where each $\boldsymbol{\Theta}_h \in \mathbb{R}^{T \times n}$ corresponds block belonging to head $h \in [H]$. Observe that $\boldsymbol{\Theta}_h$ is a right stochastic matrix (*i.e.*, rows sum up to one) for all $h \in [H]$. Thus, each row of $\boldsymbol{\Theta}_h - \boldsymbol{\Theta}_1$ sums up to zero and the matrix has at least one zero eigenvalue. Therefore,

$$\begin{aligned}
\operatorname{rank}(\boldsymbol{Z}) &\leq \operatorname{rank}(\boldsymbol{\Theta}) = \operatorname{rank}\left([\boldsymbol{\Theta}_1, \quad \boldsymbol{\Theta}_2, \quad \boldsymbol{\Theta}_3, \quad \dots, \quad \boldsymbol{\Theta}_H]\right) \\
&= \operatorname{rank}\left([\boldsymbol{\Theta}_1, \quad \boldsymbol{\Theta}_2 - \boldsymbol{\Theta}_1, \quad \boldsymbol{\Theta}_3 - \boldsymbol{\Theta}_1, \quad \dots, \quad \boldsymbol{\Theta}_H - \boldsymbol{\Theta}_1]\right) \\
&\leq \operatorname{rank}(\boldsymbol{\Theta}_1) + \operatorname{rank}(\boldsymbol{\Theta}_2 - \boldsymbol{\Theta}_1) + \cdots + \operatorname{rank}(\boldsymbol{\Theta}_H - \boldsymbol{\Theta}_1) \\
&\leq n + (n-1) + (n-1) + \cdots + (n-1) = H(n-1)+1.
\end{aligned}$$

Hence $\operatorname{rank}(\boldsymbol{Z})$ is upper bounded by $H(n-1)+1$ and this concludes the proof.

## F  PROOF OF PROPOSITION 3

We give a proof by contradiction, so let us assume $T > (n+1)m/d_{\text{out}} + m + 1$. Consider ReLU network $f$ as

$$f_{d_1, d_2, d_{\text{out}}}(\boldsymbol{x}; \boldsymbol{W}_1, \boldsymbol{b}_1, \boldsymbol{W}_2, \boldsymbol{b}_2) := \boldsymbol{W}_2^\top \operatorname{ReLU}\left(\boldsymbol{W}_1^\top \boldsymbol{x} + \boldsymbol{b}_1\right) + \boldsymbol{b}_2.$$

where $\boldsymbol{W}_1 \in \mathbb{R}^{d_2 \times d_1}, \boldsymbol{b}_1 \in \mathbb{R}^{d_2}, \boldsymbol{W}_2 \in \mathbb{R}^{d_{\text{out}} \times d_2}$, and $\boldsymbol{b}_2 \in \mathbb{R}^{d_{\text{out}}}$ are the network parameters. First, we consider $d_1 = n, d_2 = m$. This network possesses a total of $(n+1)m + (m+1)d_{\text{out}}$ parameters. Since a trivial upper bound on the number of real-valued labels a network can memorize is at most as high as the number of its parameters, for any

$$T > \frac{(n+1)m + (m+1)d_{\text{out}}}{d_{\text{out}}} = \frac{(n+1)m}{d_{\text{out}}} + m + 1$$

and inputs $\boldsymbol{x}^{(1)}, \boldsymbol{x}^{(2)}, \dots, \boldsymbol{x}^{(T)} \in \mathbb{R}^n$, there exists labels $\boldsymbol{y}^{(1)}, \boldsymbol{y}^{(2)}, \dots, \boldsymbol{y}^{(T)} \in \mathbb{R}^{d_{\text{out}}}$ such that no $f_{n,m,d_{\text{out}}}$ can memorize this dataset. In other words, for any $\boldsymbol{W}_1, \boldsymbol{W}_2, \boldsymbol{b}_1, \boldsymbol{b}_2$, there exists index $i \in [T]$ such that

$$f_{n,m,d_{\text{out}}}(\boldsymbol{x}_i; \boldsymbol{W}_1, \boldsymbol{b}_1, \boldsymbol{W}_2, \boldsymbol{b}_2) \neq \boldsymbol{y}_i.$$

Now, let us consider any arbitrary $\boldsymbol{x}^{(1)}, \boldsymbol{x}^{(2)}, \dots, \boldsymbol{x}^{(T)} \in \mathbb{R}^n$ in General Position (*i.e.*, maximal Kruskal rank of $n$), and their corresponding set of labels $\boldsymbol{y}^{(1)}, \boldsymbol{y}^{(2)}, \dots, \boldsymbol{y}^{(T)} \in \mathbb{R}^{d_{\text{out}}}$ for which no $f_{n,m,d_{\text{out}}}$ can memorize the labels. Consider the following extended input dataset:

$$\boldsymbol{c}_{\text{FCN}} := [\boldsymbol{s}_1; \boldsymbol{s}_2; \dots; \boldsymbol{s}_n; \boldsymbol{0}_{(d-n)}] \in \mathbb{R}^{(n+1)d-n}, \quad \boldsymbol{x}_{\text{FCN}}^{(t)} := [\boldsymbol{c}_{\text{FCN}}; \boldsymbol{x}^{(t)}] \in \mathbb{R}^{(n+1)d} \quad \text{for all } t \in [T], \tag{23}$$

where $\boldsymbol{s}_i$ are standard basis vectors in $\mathbb{R}^d$. Let us call $\mathcal{T}_{\text{FCN}} := \{(\boldsymbol{x}_{\text{FCN}}^{(t)}, \boldsymbol{y}^{(t)})\}_{t \in [T]}$. First, we claim that no network $f_{(n+1)d,m,d_{\text{out}}}$ can memorize $\mathcal{T}_{\text{FCN}}$. To show this, we create a mapping between any network of $f_{(n+1)d,m,d_{\text{out}}}$ to networks of $f_{n,m,d_{\text{out}}}$. For any $\boldsymbol{W}_1' \in \mathbb{R}^{m \times (n+1)d}, \boldsymbol{b}_1' \in \mathbb{R}^m, \boldsymbol{W}_2' \in \mathbb{R}^{d_{\text{out}} \times m}$, and $\boldsymbol{b}_2' \in \mathbb{R}^{d_{\text{out}}}$ define:

$$\begin{aligned}
\boldsymbol{W}_1 &:= \boldsymbol{W}_1'[:, (n+1)d - n + 1 : (n+1)d] && (\boldsymbol{W}_1 \in \mathbb{R}^{m \times n}) \\
\boldsymbol{b}_1 &:= \boldsymbol{b}_1' - \boldsymbol{W}_1'[:, 1 : (n+1)d - n]^\top \boldsymbol{c}_{\text{FCN}} && (\boldsymbol{b}_1 \in \mathbb{R}^m) \\
\boldsymbol{W}_2 &:= \boldsymbol{W}_2' && (\boldsymbol{W}_2 \in \mathbb{R}^{d_{\text{out}} \times m}) \\
\boldsymbol{b}_2 &:= \boldsymbol{b}_2', && (\boldsymbol{b}_2 \in \mathbb{R}^{d_{\text{out}}})
\end{aligned}$$

where $\boldsymbol{W}_1'[:, a : b]$ is the sub-block of matrix $\boldsymbol{W}_1'$ selected by columns $a$ to $b$. With the above assignment, we get:

$$f_{(n+1)d,m,d_{\text{out}}}(\boldsymbol{x}_{\text{FCN}}^{(t)}; \boldsymbol{W}_1', \boldsymbol{b}_1', \boldsymbol{W}_2', \boldsymbol{b}_2') = f_{n,m,d_{\text{out}}}(\boldsymbol{x}^{(t)}; \boldsymbol{W}_1, \boldsymbol{b}_1, \boldsymbol{W}_2, \boldsymbol{b}_2),$$

for all $t \in [T]$. Therefore, since no ReLU network $f_{n,m,d_{\text{out}}}$ is able to memorize $\{(\boldsymbol{x}^{(t)}, \boldsymbol{y}^{(t)})\}_{t \in [T]}$, no ReLU network $f_{(n+1)d,m,d_{\text{out}}}$ is able to memorize $\mathcal{T}_{\text{FCN}}$ either.

So far, we have found a dataset $\mathcal{T}_{\text{FCN}}$ that a two-layer ReLU network with $m$ hidden neurons can not memorize. Next, we prove that $\mathcal{T}_{\text{FCN}}$ can be derived from a valid dataset $\mathcal{T} = \left\{ \left( \boldsymbol{E}^{(t)}, \boldsymbol{e}^{(t)}, \boldsymbol{y}^{(t)} \right) \right\}_{t=1}^T$, which satisfies both Assumptions 1 and 2. To observe this, consider the following inputs:

$$\boldsymbol{E}^{(t)} := [\boldsymbol{s}_1^\top; \boldsymbol{s}_2^\top; \ldots, \boldsymbol{s}_n^\top]$$

$$\boldsymbol{e}^{(t)} := [\boldsymbol{0}_{(d-n)}; \boldsymbol{x}^{(t)}],$$

and labels $\boldsymbol{y}^{(t)}$ the same as $\mathcal{T}_{\text{FCN}}$. Then, we get

$$\boldsymbol{x}_{\text{FCN}}^{(t)} = [\boldsymbol{e}_1^{(t)}; \ldots; \boldsymbol{e}_n^{(t)}; \boldsymbol{e}^{(t)}].$$

So, what is left is to prove that $\mathcal{T}$ satisfies Assumptions 1 and 2. First, $\{\boldsymbol{e}^{(t)}\}_{t\in[T]}$ has the same Kruskal rank as $\{\boldsymbol{x}^{(t)}\}_{t\in[T]}$, which is $n$, so Assumption 1 is satisfied. Second, each $\boldsymbol{E}^{(t)}$ is full-rank since each row $i \in [n]$ is the $i$-th standard basis for all $t \in [T]$.

Therefore, we have found a dataset $\mathcal{T}$ with size $T > (n+1)m/d_{\text{out}} + m + 1$ that no ReLU network with $m$ hidden dimensions can memorize, and this completes the proof.

## G  PROOF OF PROPOSITION 4

First, we derive the output tokens $\boldsymbol{E}'^{(t)}, \boldsymbol{e}'^{(t)}$ as a function of input tokens. We omit the head subscript since we have only one head. Since $\boldsymbol{W}_K = \boldsymbol{W}_Q = \boldsymbol{0}$, we have $\boldsymbol{\alpha}^{(t)} = \boldsymbol{0}$, and consequently $\boldsymbol{\theta}^{(t)} = \frac{1}{n}\boldsymbol{1}$. Therefore, the attention head only takes the average of each token, *i.e.*,

$$\bar{\boldsymbol{e}} := \frac{1}{n}\left( \boldsymbol{e}_1^{(t)} + \boldsymbol{e}_2^{(t)} + \cdots + \boldsymbol{e}_n^{(t)} \right).$$

Now, since the Attention head also has a skip connection, the output of each token becomes:

$$\boldsymbol{e}'^{(t)} = \boldsymbol{e}^{(t)} + \boldsymbol{o}^{(t)} = \boldsymbol{e}^{(t)} + \bar{\boldsymbol{e}}, \tag{24}$$

$$\boldsymbol{e}_i'^{(t)} := \boldsymbol{e}_i^{(t)} + \boldsymbol{o}_i^{(t)} = \boldsymbol{e}_i^{(t)} + \bar{\boldsymbol{e}} \quad \text{for all } i \in [n] \tag{25}$$

First, since we have $\boldsymbol{e}'^{(t)} = \tilde{\boldsymbol{e}}^{(t)}$ in Assumption 3, this means that the set $\{\boldsymbol{e}'^{(t)}\}_{t=1}^T$ has a Kruskal Rank at least $n$, and Assumption 1 holds for $\mathcal{T}'$. Next, since the $\boldsymbol{E}^{(t)}$ are full-rank, adding the average of tokens to $\boldsymbol{E}^{(t)}$ preserves the rank (Lemma 2), so $\boldsymbol{E}'^{(t)}$ are also full-rank and Assumption 2 is satisfied for $\mathcal{T}'$.

## H  SOFTMAX SATURATION AND OPTIMIZATION-BASED MEMORIZATION

In the proof of Theorem 1, we took advantage of the saturation properties of Softmax. We generate a dataset of size $T$ with a shared context (as in Section 5.2) with $d = 64$, $n = 32$, and discrete labels $y^{(t)} \in \mathcal{U}([100])$ drawn *iid* from a uniform distribution. We test with two different head sizes $H = 4$ and $H = 8$. For each head size, we first empirically find the maximal number of data points the model can perfectly memorize using gradient-based optimization. We find for $H = 4$, this number is around $T = 550$, and for $H = 8$ around $T = 1300$. We train both models for $300{,}000$ optimization steps. We call a head *saturated* for an example if the query token has one softmax coefficient $> 0.99$ to some context token in that head, *i.e.*, head $h \in [H]$ is saturated for example $t \in [T]$ if $\boldsymbol{\theta}_h^{(t)}$ has at least one entry $> 0.99$.

The results are depicted in Figure 3. Interestingly, a significant number of heads are saturated for the majority of examples, while almost all examples have at least one corresponding non-saturated head. This is in line with the intuition we provided in the proof of Theorem 1, where our proof relied on designating a set of examples to one head and saturating that head for the rest of the examples.

## I  UPPER BOUND ON THE RANK OF MATRIX Z WITH STRONGER ASSUMPTIONS

In Proposition 2 we provided an upper bound on the rank of $\boldsymbol{Z}$, which matches the lower bound of Proposition 1. One might wonder whether under stronger assumptions (and perhaps far from reality), a full-rank $\boldsymbol{Z}$ with rank $Hd$ is achievable or not. Here, we prove that even the "General Position" on

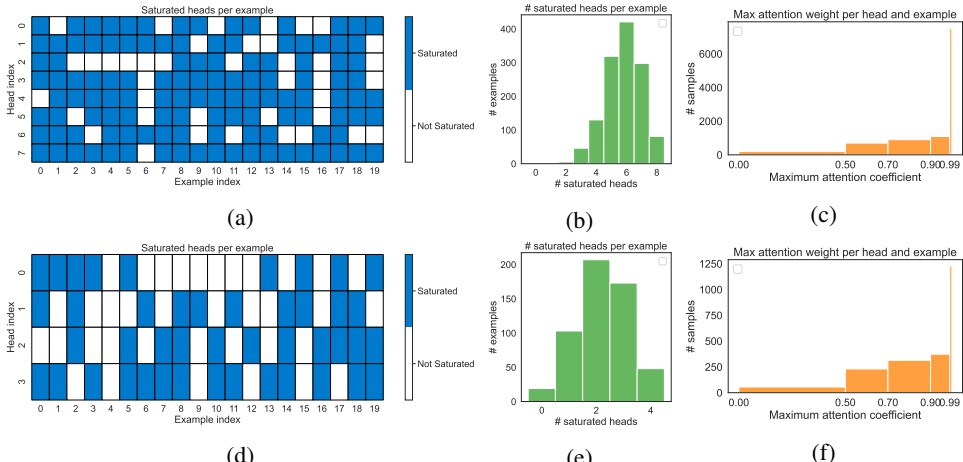

Figure 3: Testing the saturation property of Softmax on synthetic data. On the top row, we have $H = 8$, and on the bottom row $H = 4$. (a) and (d) show whether softmax in each head is saturated for the first 20 examples of the dataset, results are qualitatively similar for the rest of the examples. (b) and (e) show the histogram of the number of saturated heads for each example across the dataset ($H \times T$ in total). (c) and (f) show the histogram of the maximum softmax coefficient for each head and example across all the datasets ($H \times T$ in total). All the figures suggest that with perfect memorization, the majority of heads become saturated across the examples, with almost all examples having at least one non-saturated head.

*all* tokens assumption is not enough, and there exists a very simple assumption that limits the rank of $\boldsymbol{Z}$ to $H(d-1) + 1$.

**Proposition 5.** *Assume that we have a multi-head attention layer $\mathcal{A}$ with $H$ heads, embedding dimensions $d = d_h = d$, and $d_v = 1$. Let $\mathcal{T} = \left\{ \left( \boldsymbol{E}^{(t)}, \boldsymbol{e}^{(t)}, \boldsymbol{y}^{(t)} \right) \right\}_{t=1}^{T}$ be a training set with context size $n < d$. If all context vectors satisfy $\mathbf{1}^\top \boldsymbol{e}_i^{(t)} = C$ for all $t \in [T]$, $i \in [n]$ and some universal scalar $C \in \mathbb{R}$, then for any set of weights $\{(\boldsymbol{W}_{Kh}, \boldsymbol{W}_{Qh})\}_{h=1}^{H}$, we have*
$$\operatorname{rank}(\boldsymbol{Z}) \le H(d-1) + 1.$$

Proposition 5 gives an upper bound on the rank of $\boldsymbol{Z}$ when individual elements of each context token sum up to a constant $C$. Note that all the context vectors can be in general position and still satisfy this condition. Moreover, this upper bound puts *no assumption* on the query vectors. Hence even with the strongest assumptions, obtaining a full-rank matrix $\boldsymbol{Z}$ is out of reach. However, whether stronger assumptions can maintain dependence on $d$ instead of $n$ (*i.e.*, give $H(d-1)+1$ memorization lower bound instead of $H(r-1)+1$) is still an open question for future work.

*Proof of Proposition 5.* First, note that since $\mathbf{1}^\top \boldsymbol{e}_i^{(t)} = C$, and $\boldsymbol{\theta}_i^{(t)}$ are convex combinations we have:
$$\mathbf{1}^\top \boldsymbol{z}_h^{(t)} = \mathbf{1}^\top \boldsymbol{E}(t)^\top \boldsymbol{\theta}_h^{(t)} = C \mathbf{1}^\top \boldsymbol{\theta}_h^{(t)} = C \quad \text{for all } t \in [T].$$
With some abuse of notation, let us write the feature matrix $\boldsymbol{Z} \in \mathbb{R}^{T \times Hd}$ as follows:
$$\boldsymbol{Z} = [\boldsymbol{Z}_1, \quad \boldsymbol{Z}_2, \quad \ldots, \quad \boldsymbol{Z}_H],$$
where each $\boldsymbol{Z}_h \in \mathbb{R}^{T \times d}$ is a sub-block of $\boldsymbol{Z}$ corresponding to the $h$-th head for all $h \in [H]$. Then, since each row of $\boldsymbol{Z}_h$ (*i.e.*, $\boldsymbol{z}_h^{(t)}$) sums up to $C$, with an argument similar to the proof of Proposition 2 we get:
$$\begin{aligned}
\operatorname{rank}(\boldsymbol{Z}) &= \operatorname{rank}\left( [\boldsymbol{Z}_1, \quad \boldsymbol{Z}_2, \quad \boldsymbol{Z}_3, \quad \ldots, \quad \boldsymbol{Z}_H] \right) \\
&= \operatorname{rank}\left( [\boldsymbol{Z}_1, \quad \boldsymbol{Z}_2 - \boldsymbol{Z}_1, \quad \boldsymbol{Z}_3 - \boldsymbol{Z}_1, \quad \ldots, \quad \boldsymbol{Z}_H - \boldsymbol{Z}_1] \right) \\
&\le \operatorname{rank}(\boldsymbol{Z}_1) + \operatorname{rank}(\boldsymbol{Z}_2 - \boldsymbol{Z}_1) + \cdots + \operatorname{rank}(\boldsymbol{Z}_H - \boldsymbol{Z}_1) \\
&\le d + (d-1) + (d-1) + \cdots + (d-1) \\
&= H(d-1) + 1.
\end{aligned}$$

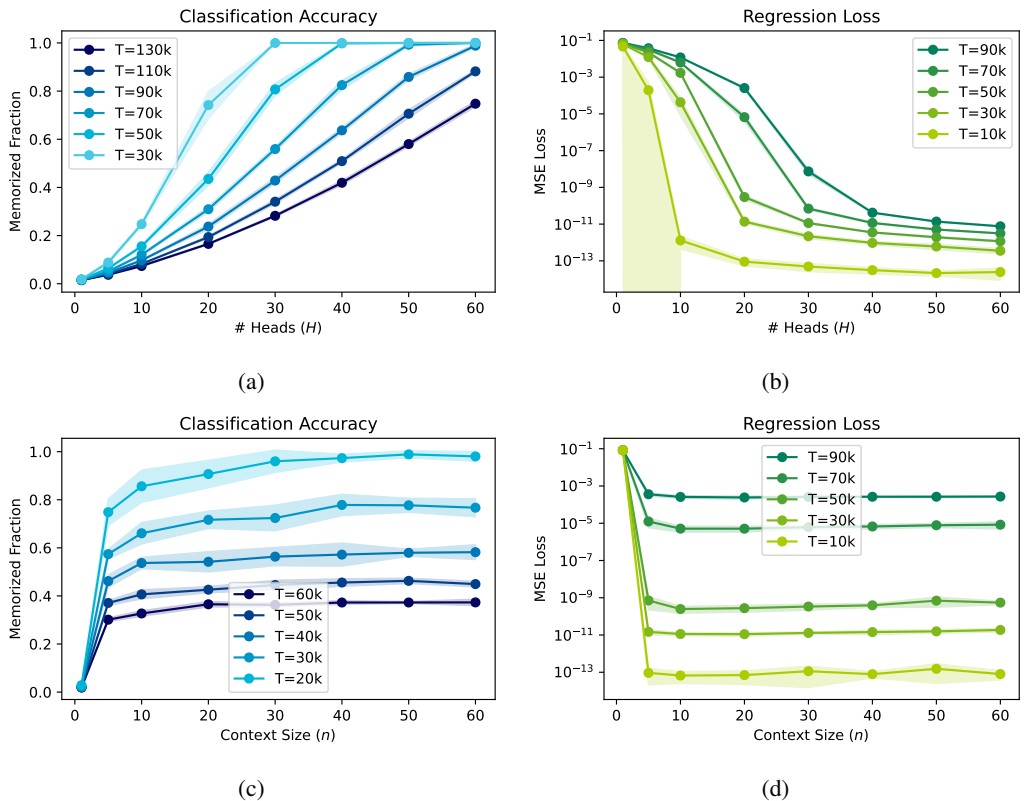

(a)          (b)

(c)          (d)

Figure 4: Testing memorization under general position assumption for both context and query vectors. While the linearity in $H$ still remains, the linearity in $n$ no longer strictly holds, suggesting $H(d-1)+1$ to be a better alternative under general position assumption.

$\square$

## I.1 Synthetic experiments under general position assumption

We also run synthetic experiments under the general position assumption on *all* input tokens. We consider a similar setting to Section 5.2, but we also generate context vectors from a uniform distribution as well, *i.e.*

$$\boldsymbol{x}_1^{(t)}, \boldsymbol{x}_2^{(t)}, \ldots, \boldsymbol{x}_n^{(t)}, \boldsymbol{x}^{(t)} \overset{iid}{\sim} \mathcal{U}(0,1)^{64} \quad \text{for all } t \in [T].$$

This way, all tokens (*i.e.*, context and query) inside, and across all examples would satisfy the general position assumption. Preliminary results in Figure 4 suggest that the linearity in $H$ still remains, while the linearity in $n$ is weakened. We conjecture that a bound with complexity $O(Hd)$ would hold, which could be interesting for future works. Yet, as we showed in Section 5.1, from a practical point of view, the General Position assumption might be too strong to be realistic.

## J Testing Assumptions on Natural Language Tasks

We further test Assumptions 2 and 1 on BERT Devlin et al. (2019), and GPT2 Radford et al. (2019) on the task of pre-training on Wikipedia Foundation. Following the setting introduced in Section 5.1, we examine the embedding/first layer of trained/untrained GPT2/BERT.

- **BERT:** We follow a setting similar to the pre-training of BERT. For each example, an input of the format *"[CLS] Sentence1 [SEP] Sentence2"* is formed, where *Sentence 1* and *Sentence 2* are two pieces of text from the Wikipedia corpus. Here *Sentence 2* is the sentence after *Sentence 1* with probability $0.5$ and randomly chosen from the corpus with probability

Table 3: Testing assumptions on Wikipedia for BERT and GPT2. L0 refers to the embedding layer, and L1 refers to the output of the first layer.

| Model | GPT2 | | | | BERT | | | |
|---|---|---|---|---|---|---|---|---|
| | Random | | Trained | | Random | | Trained | |
| | L0 | L1 | L0 | L1 | L0 | L1 | L0 | L1 |
| General Position | × | × | × | × | × | × | × | × |
| Assumption 1 | × | ✓ | × | ✓ | × | × | × | × |
| Assumption 2 | ✓ | ✓ | ✓ | × | ✓ | ✓ | ✓ | ✓ |

0.5. The "[CLS]" token's output predicts whether *Sentence 2* follows the first sentence or not. We consider the "[CLS]" token as the query token $e^{(t)}$, and the whole input as the context $E^{(t)}$ for all $t \in [T]$. The positional encoding is sinusoidal positional encoding.

- **GPT2:** This model is trained autoregressively to predict the next token in each piece of text. We consider last-token prediction from the context. That is, we set the query $e^{(t)}$ to be the last token of an example, and the context $E^{(t)}$ to be the whole input example. In contrast to BERT, The positional encoding in this model is learned instead of fixed.

The results of testing Assumptions are shown in Table 3. While linear independence assumptions are not well-suited for discrete data, we observe that most assumptions still hold. For Assumption 2, the sinusoidal positional encodings in BERT preserve the linear independence for all cases. In GPT2, we observe that the learned positional encodings in pre-trained GPT2 have a lower rank and are no longer linearly independent. For Assumption 1, the GPT2 model already satisfies it after one layer. In the case of BERT, the $n/d$ ratio is $512/1024$, and while the Kruskal Rank of query vectors is below 512 (violation of Assumption 1), we observe that the Kruskal Rank is high enough (291 for the case of trained and 349 for the case of random BERT). Hence, the result of Remark 1 still yields an acceptable lower bound.

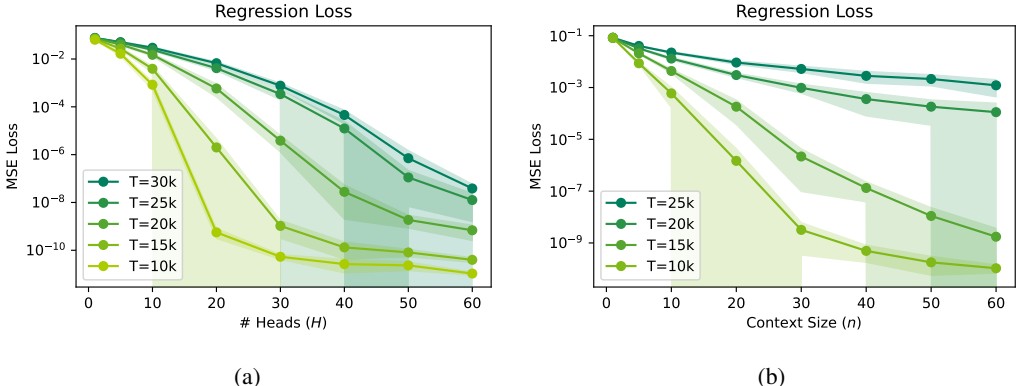

Figure 5: Testing memorization on regression task with a similar setting as Figure 2, except for real-valued labels. The same results and conclusions in terms of monotonicity with $H$ and $n$ hold true for regression. The variance in regression plots is magnified in some cases due to the log scale of the plot.

