# OpenReview forum: "Memorization Capacity of Multi-Head Attention in Transformers"
_ICLR.cc/2024/Conference — ICLR 2024 spotlight_

### Official Review · Reviewer_oLtn · 2023-11-01

**Soundness:** 4 excellent
**Presentation:** 4 excellent
**Contribution:** 3 good
**Rating:** 8
**Confidence:** 3

**Summary:**

The authors prove a lower bound on the memorization capacity of a multi-head attention layer under a new set of input-data assumptions that relax General Position assumptions previously used in similar work. The main result is that memorization increases linearly with the number of heads, monotonically with the context size, and monotonically with the head dimension as long as it is smaller than the context size. The result validates some of the common practical choices of hyperparameters. The authors verify their data assumptions empirically by checking for linear independence of context vectors and approximately testing the Kruskal Rank. They also present a synthetic experiment that confirms their theoretical findings.

**Strengths:**

The paper presents an important result on the memorization capacity of a multi-head attention layer using an original set of data assumptions that are shown to hold in practice. The result is verified through a properly designed experiment. The paper is well written and structured, and the logic of the argument is easy to follow. The figures are of high quality and they help to get the message across. While I haven't checked the proof in detail, the main steps seem to be sound.

**Weaknesses:**

It would be great to provide code for your synthetic experiments and input-data assumption validation to increase reproducibility.

**Questions:**

How good is the approximation of the Kruskal Rank computation?

How did you come up with this particular set of data assumptions? Was this dictated by the proof technique you wanted to apply?

Can your proof technique be applied to prove a similar result beyond a single attention layer or is there some fundamental obstacle?

---

> ### Author Response · Authors · 2023-11-16
>
> We thank the reviewer for their positive comments and encouraging feedback.
>
> **Q1: It would be great to provide code for your synthetic experiments and input-data assumption validation to increase reproducibility.**
>
> Ensuring the reproducibility of our research is of utmost importance to us, and we have included the code for all of our experiments in the supplementary material with instructions on how to reproduce the experiments. We will also publicly release the code in the camera-ready version of our paper.
>
> **Q2: How good is the approximation of the Kruskal Rank computation?**
>
> As mentioned in section 5.1, computing Kruskal Rank is an NP-Hard problem. As such, any polynomial-time algorithm is suboptimal for such computation. The randomized algorithm used in our paper (which picks random subsets and tests matrix rank) keeps a trade-off between computational time and testing accuracy.
>
> **Q3: How did you come up with this particular set of data assumptions? Was this dictated by the proof technique you wanted to apply?**
>
> Our Assumption 1 is inspired by the "General Position" assumption found in several prior works on ReLU FCN [1, 6]. However, as our experiments indicate (Table 1), this assumption is often not met in practice. This led us to propose the Kruskal rank assumption as a relaxation to the general position (i.e., it subsumes the general position assumption as a special case), which we also found to better-suite practical settings (specifically kurskal rank larger than context size $n$ in Table 2). Moreover, as mentioned in section 4.4, the positional encoding used in transformers is the primary motivation behind Assumption 2.
>
> **Q4: Can your proof technique be applied to prove a similar result beyond a single attention layer or is there some fundamental obstacle?**
>
> While extending our proof technique to multi-layer attention networks requires further exploration, we anticipate certain proof insights from current results to stay relevant for this extension. (1) In our construction, we group data points and assign them to individual heads, a strategy that may be extended to multiple layers. For example, previous studies on multi-layer ReLU FCNs (cf [1]) have employed multi-step grouping to establish bounds for multi-layer ReLU networks, providing a potential avenue for adapting and applying similar ideas to attention networks. (2) In our proof technique, we also exploited the saturation property of softmax to allow grouping data per head without distorting output for other data points. We anticipate such a construction to stay relevant for extension beyond one layer.
>
> In closing, we mention that a preliminary step along these directions is taken in Proposition 4 (where a constructed attention layer as the first layer helps satisfy assumptions for the subsequent layer). We eagerly anticipate seeing further improvements and extensions of our theorem to multi-layer attention models.
>
> [1] C. Yun et al. Small ReLU networks are powerful memorizers: a tight analysis of memorization capacity, NeurIPS 2019

---

### Official Review · Reviewer_mRPx · 2023-11-07

**Soundness:** 2 fair
**Presentation:** 3 good
**Contribution:** 2 fair
**Rating:** 8
**Confidence:** 2

**Summary:**

this paper investigates the memorization abilities of MHA of transformers with theorectical analysis. Based on two new proposed assumptions, the authors find a lower bound on the memorization capacity of attention layers. The paper tested the rationality of the assumptions and validate the theorectial findings with synthetic experiments.

**Strengths:**

1. The assumptions in this paper are more relaxed The authors  verified the rationality of the assumptions on real data.
2. The exploration of memorization capacity of transformers is meaningful for more advanced go-to architecture, while the memorization abilities of attention modules is quite interesting.
3. The paper is well-written.

**Weaknesses:**

1. One of my main concern is the illustration or definition of "memorization" in this paper. The inputs of attention include both the key matrix and the query vector. In a common understanding, attention plays a role to capture knowledge from the context according to the "attention" on other tokens for each token.  So what does attention memorize? I think the paper should make it clearer before or after the theorectical analysis, or even verify the memorized knowledge with some visualization.
2. Analysis and comparison with existing works are insufficient, for example,  <<Transformer Feed-Forward Layers Are Key-Value Memories>>. How are the memories different between this paper and the reference? There are also some works about transformer   interpretability and model editing. Do the observations in these related works support the findings in this paper? I think the authors should give more analysis, rather than demonstrating the their propsed theorem only.

**Questions:**

refer to the weakness.

---

> ### Author Response · Authors · 2023-11-16
>
> We thank the reviewer for their constructive comments.
>
> **Q1: One of my main concern is the illustration or definition of "memorization" in this paper. The inputs of attention include both the key matrix and the query vector. In a common understanding, attention plays a role to capture knowledge from the context according to the "attention" on other tokens for each token. So what does attention memorize? I think the paper should make it clearer before or after the theorectical analysis, or even verify the memorized knowledge with some visualization.**
>
> In response to the concern about the definition of "memorization", we adopt a classical definition applicable to any parameterizable function $f$ and a set of $T$ input/output pairs $x_i$, $y_i$. This definition aligns with established literature on both ReLU networks and Transformers [3, 4, 5, 6, 7]. Our focus in this work is on Attention-only models, isolating them from the well-studied ReLU networks to explore their capabilities in the absence of ReLU activation.
>
> While the common belief attributes the success of attention to its ability to capture knowledge from various parts of the input, we argue that this is not the sole reason behind its efficacy. We explicitly recognize the utility of attention in capturing input context information in Section 4.4 and Proposition 4, where we demonstrate its effectiveness in token-mixing (or contextualization). Moreover, in our proof technique, we show that we can partition input data into subsets and make each head of the Multi-Head attention to be responsible for memorizing only one subset. This responsibility is not in the form of "capturing knowledge from the context", but rather, exploiting the context to build a desired convex combination output vector (see Appendix B for complete proof). Moreover, we do not impose any explicit assumption on the relationship between the input and label, and our theorem works for any arbitrary label assigned to an input datapoint, therefore the notion of "knowledge" is not well-defined. Finally, there have been recent empirical observations that show merely extending the transformer's input context with *constant* tokens can enhance the performance of transformers (e.g., [1, 2]). Incorporating such constant tokens enhances the performance without adding any "knowledge" to the context.
>
> **Q2: Analysis and comparison with existing works are insufficient, for example, <<Transformer Feed-Forward Layers Are Key-Value Memories>>. How are the memories different between this paper and the reference? There are also some works about transformer interpretability and model editing. Do the observations in these related works support the findings in this paper? I think the authors should give more analysis, rather than demonstrating the their propsed theorem only.**
>
> Thanks for bringing this paper to our attention. First, we would like to highlight the difference between "neural memory" definition, and the "memorization" definition used in our work. In neural memory, typically the differentiable weights are seen as memory units, and the neural network is seen as a function that utilizes/modifies these weights to produce an answer. In our work, however, memorization is defined as the network being able to memorize (i.e., interpolate) the output $y$ for each corresponding input $x$. In memory units, the network has a specific structure such that some weights have specific interpretations as memory units, while in memorization, the goal is to achieve an output given input with the smallest possible parameter count. Second, the work you mentioned focuses on the fully-connected ReLU part of a Transformer, and interprets its first layer weights as "memory keys", and its second layer as "memory values". **While it is interesting to study the role of ReLU FCN in transformers, we take an orthogonal approach and focus on isolating the attention part of the transformer.** As a result, the focus and scope of our paper differ from the work you mentioned, making direct comparisons challenging. In the new revision, we updated our manuscript with a footnote to clarify the distinction between concepts.
>
> If you have additional specific papers in mind that you believe would enhance the discussion, we are open to incorporating them into our analysis to further enhance the exposition of our work and its relation to existing literature.
>
> [1] S. Goyal et al. Think before you speak: Training Language Models With Pause Tokens, 2023
>
> [2] T. Darcet et al. Vision Transformers Need Registers, 2023
>
> [3] R. Vershynin. Memory capacity of neural networks with threshold and rectified linear unit activations. SIAM J., 2020.
>
> [4] G. Vardi, et al. On the optimal memorization power of reLU neural networks. ICLR, 2022.
>
> [5] C. Yun, et al. Small relu networks are powerful memorizers: a tight analysis of memorization capacity. NeurIPS, 2019.

---

> > ### Author Response · Authors · 2023-11-16
> > **Official Comment by Authors (Cont.)**
> >
> > [6] S. Bubeck, et al. Network size and size of the weights in memorization with two-layers neural networks, NeurIPS, 2020.
> >
> > [7] J. Kim et al. Provable Memorization Capacity of Transformers, ICLR 2023.

---

> > ### Comment · Reviewer_mRPx · 2023-11-22
> >
> > The authors addressed my concerns and i will raise my rating.

---

### Official Review · Reviewer_SUAq · 2023-11-08

**Soundness:** 3 good
**Presentation:** 3 good
**Contribution:** 3 good
**Rating:** 8
**Confidence:** 3

**Summary:**

This paper proposes a lower bound on the memorization capacity of MHA layers under a rather mild linear independence assumption. Based on two input-data assumptions, the authors theoretically and experimentally prove that (1) When fixing dimension d, increasing the number of heads H improves memorization. (2) When further fixing the number of heads H, increasing the context size n improves memorization. (3) When fixing d, n, increasing dh only helps up to dh < n, there is no memorization gain beyond that.

**Strengths:**

1.	The paper is well-organized and the proof makes sense.
2.	The two input-data assumptions are milder than the General Position assumptions. Although it is impossible to fully verify its generalizability, the author demonstrated the reasonableness of the assumptions through sampling testing, which interests me.
3.	The conclusion “When fixing d, n, increasing dh only helps up to dh < n, and there is no memorization gain beyond that” is enlightening and I believe it can bring more valuable thinking and discussion to the community.

**Weaknesses:**

1.	It might be significantly different between the image patch tokens (ViT) and the language tokens. Can the author's experimental verification of those assumptions be verified on NLP tasks?

**Questions:**

1.	The authors demonstrate that Assumption 2 typically holds in practice due to positional encoding. Does the assumption still hold when the positional encoding is learnable?

---

> ### Author Response · Authors · 2023-11-16
>
> We thank the reviewer for their positive comments and encouraging feedback.
>
> **Q1: It might be significantly different between the image patch tokens (ViT) and the language tokens. Can the author's experimental verification of those assumptions be verified on NLP tasks?**
>
> As mentioned in Section 4, our linear independence assumptions are primarily best-suited for continuous input data such as images, hence our focus on ViT and ImageNet. For completeness, we have included experiments on GPT-2 and BERT in Appendix J. We find that randomly initialized GPT-2 satisfies both assumptions and Theorem 1 is applicable in this setting. Moreover, a randomly initialized BERT satisfies Assumption 2. For BERT, while it appears to violate Assumption 1, our theory provides a (weaker) bound on memorization as discussed in Remark 1.
>
> **Q2: The authors demonstrate that Assumption 2 typically holds in practice due to positional encoding. Does the assumption still hold when the positional encoding is learnable?**
>
> Yes. When positional encodings are learnable, they become a part of the designed parameter set for memorization. One simple such design is setting the positional encodings randomly from a continuous distribution, such as uniform or Gaussian. This also agrees with practice when initializing neural network weights. This random initialization ensures that each entry is independent. Given that the number of positions ($n$) is less than the dimensionality of the encoding ($d$), the positional encoding vectors $\text{pos}_1, \text{pos}_2, \dots, \text{pos}_n \in \mathbb{R}^d$ are almost surely linearly independent (i.e., are linearly independent with probability one). This is because for each $\text{pos}_i$, the set of possible vectors that make $\text{pos}_i$ a linear combination of $\text{pos}_1, \text{pos}_2, ..., \text{pos}_j$ (for $j<i$) has measure zero. Consequently, these learnable positional encodings provably help satisfy Assumption 2 (like the sinusoidal positional encodings). However, for a trained network, no definitive statement can be made. Learnable positional encodings are subject to change during training, and their linear independence may evolve in either direction. Therefore, adherence to Assumption 2 cannot be theoretically guaranteed once the network has undergone training. But, again, since memorization capacity asks for *existence* of parameters memorizing the data, positional encoding becomes a part of the whole set of parameters, which can be "designed" by our proof.

---

> > ### Comment · Reviewer_SUAq · 2023-11-21
> > **Response to Authors Rebuttal**
> >
> > I appreciate the authors' addressing of my questions. Now it's clear to me that how well the Assumption 2 holds for different positional encoding strategies. Thank you again and good luck.

---

### Official Review · Reviewer_6Bvj · 2023-11-10

**Soundness:** 3 good
**Presentation:** 2 fair
**Contribution:** 3 good
**Rating:** 6
**Confidence:** 3

**Summary:**

This paper explores the memorization capabilities of multi-head attention mechanisms, a core component of transformer architectures widely used for language and vision tasks. The study aims to understand how effectively these mechanisms can remember example sequences, depending on the number of attention heads and the lengths of the sequences. The research introduces new assumptions about the linear independence of input data, which differ from previous models, and under these conditions, it's shown that an attention layer with a certain number of heads and parameters can memorize a proportional number of examples. The paper further discusses how different attention heads process different sequences and how the softmax function contributes to this capability. These theoretical findings are supported by experimental results using synthetic data.

**Strengths:**

- The paper makes theoretical contributions by exploring the memorization capacity of transformers, an area that is not yet fully understood. This contributes to a deeper understanding of transformer architectures.
- The paper introduces new assumptions about the linear independence of input data, distinct from commonly used assumptions. This novel approach provides a fresh perspective on analyzing transformer models.
- The findings are validated through experiments on synthetic data. This empirical approach strengthens the theoretical claims made in the paper.
- The paper includes a detailed analysis of assumptions, theoretical proofs, and experimental validation, making it a comprehensive study.

**Weaknesses:**

- Limited Empirical Testing: While the paper includes synthetic experiments, real-world data experiments might be needed to fully understand the practical implications of the findings.
- Focus on Single-Layer MHA Module: The study primarily focuses on a single-layer Multi-head Attention (MHA) module. Expanding the analysis to multi-layered architectures could provide more comprehensive insights.
- Potential for Broader Impact Analysis: The paper could benefit from a more in-depth discussion on how these findings impact current transformer-based models in various applications, like natural language processing or computer vision.

**Questions:**

- Could you explain the rationale behind the specific assumptions made regarding the linear independence of input data? How do these assumptions align with real-world data scenarios in transformer applications?
- How do your findings contribute to the ongoing discussion in machine learning between model generalization and memorization, particularly in the context of overfitting?
- How do your derivation connect to the Hopfield Network as one classical memory network that has recently been proven to be connected to the Transformer network [1]?

[1] Ramsauer, Hubert, et al. "Hopfield networks is all you need." arXiv preprint arXiv:2008.02217 (2020).

---

> ### Author Response · Authors · 2023-11-16
>
> We thank the reviewer for their positive comments. We appreciate your time.
>
> **Q1: Limited Empirical Testing: While the paper includes synthetic experiments, real-world data experiments might be needed to fully understand the practical implications of the findings.**
>
> We would like to point out that we have indeed verified our assumptions on real-world models and datasets, as discussed in Section 5.1 of the paper. For testing memorization capacity on real-world data, the issue lies in the nature of memorization question, which only asks for the **existence** of a solution. There is no apriori easy way to check this. As we discussed in Appendix A, the nature of gradient-based optimization (e.g., using Adam optimizer) as a proxy for quantifying memorization presents a significant challenge in terms of computational resources. For instance, even for a small-scale model and synthetic dataset with a dimension of 64, we had to run for 300,000 optimization steps to mitigate optimization-related confounding limiting factors in our plot. In conclusion, given the existential nature of the question of memorization capacity, the more relevant question is that of verifying the validity of our data assumptions. As mentioned, we have done so in real-world experiments as suggested by the reviewer.
>
>
> **Q2: Focus on Single-Layer MHA Module: The study primarily focuses on a single-layer Multi-head Attention (MHA) module. Expanding the analysis to multi-layered architectures could provide more comprehensive insights.**
>
> We would like to draw attention to the progression of results for memorization in fully-connected networks. The first results appeared in 1988[1] for only a single hidden layer, and with a strong “general position” assumption (which, in our paper, we showed is highly unrealistic in our setup and offered a relaxed version based on Kruskal Rank). Only decades later, results for multi-layer ReLU networks have been proposed and various assumptions lifted[2, 3, 4], yet, the exact memorization capacity of MLPs of depth $\geq 3$ is still not fully known (only known up to logarithmic and constant factor).
>
> Here we take an analogous first step for attention models. Thus, we expect our analysis to inspire further investigations and extensions to multi-layer architectures.
>
> **Q3: Potential for Broader Impact Analysis: The paper could benefit from a more in-depth discussion on how these findings impact current transformer-based models in various applications, like natural language processing or computer vision.**
>
> The study of the memorization capacity of neural networks is a classical question in machine learning [1] and has strong ties to other classical notions, such as VC and shattering dimensions [3]. At its core, the study of memorization seeks to understand what is the minimum size (in terms of the number of parameters, but also width/depth, etc) as well as the optimal arrangement (e.g., pyramidal vs expanding networks, or head dimension as a function of embedding dimension and context length) of architecture in order to be able to memorize the training set. Answers to these provide guidance and insights on practical architecture-level design choices in terms of efficiency. At the same time, quantifying memorization also has implications in terms of generalization, and privacy [5]. Given these, we believe our results hold significance particularly since the topic is underexplored from a theoretical perspective for the transformer architecture. We anticipate this line of work can inspire theory-informed standardizations and improvements on current architectures. For instance, our analysis shows that using head dimension $d_h > n$ (but less than $d$) results in no loss in memory capacity. Having said these, we acknowledge that this is only the beginning towards more immediate relevant implications for SOTA transformers, we anticipate (and are also working ourselves on) to see extensions in terms of architectural assumptions as well as further investigations of the closely related questions of optimization.
>
>
> **Q4: Could you explain the rationale behind the specific assumptions made regarding the linear independence of input data? How do these assumptions align with real-world data scenarios in transformer applications?**
>
> Our Assumption 1 is inspired by the "General Position" assumption found in several prior works on ReLU FCN [1, 6]. However, as our experiments indicate (Table 1), this assumption is often not met in practice. This led us to propose the Kruskal rank assumption as a relaxation to the general position (i.e., it subsumes the general position assumption as a special case), which we also found to better-suite practical settings (specifically kurskal rank larger than context size $n$ in Table 2). Moreover, as mentioned in section 4.4, the positional encoding used in transformers is the primary motivation behind Assumption 2.

---

> > ### Author Response · Authors · 2023-11-16
> > **Official Comment by Authors (Cont.)**
> >
> > **Q5: How do your findings contribute to the ongoing discussion in machine learning between model generalization and memorization, particularly in the context of overfitting?**
> >
> > While our work does not directly consider generalization, we believe precise quantification of the memorization ability of models on real-world datasets provides significant insight into generalization (e.g., measuring how much of the success of foundation models can be attributed to memorization rather than generalization). We take a step towards better such quantification by (1) refining common assumptions on the data towards being more realistic (Assumptions 1 and 2) and (2) proposing a lower bound on the memorization with such assumptions (Theorem 1). Please also see our response to Q3.
> >
> > **Q6: How do your derivation connect to the Hopfield Network as one classical memory network that has recently been proven to be connected to the Transformer network?**
> >
> > Thanks for providing the reference! We will discuss this work in our later version.
> >
> > This study aligns with our research objectives. Specifically, the authors explore the memorization capacity of a continuous Hopfield network, where a one-step concave-convex procedure (CCCP) update is equivalent to a single-head attention layer. In contrast, our focus is on analyzing the memorization capacity of a multi-head attention layer.
> >
> > Nevertheless, it is crucial to underscore two primary distinctions.
> >
> > 1) Their definition of memorization is that the retrieved pattern is $\epsilon$-close to a fixed point of the energy function. However, this fixed point may not necessarily correspond to a data point from the training set. It could be a point near the data point, particularly if the patterns are well separated, or even metastable points (e.g., a superposition of multiple or all data points). In contrast, we study the exact memorization (no tolerance of $\epsilon$ error) of training data points.
> >
> > 2) Their theoretical result regarding the number of memorized data points (exponential w.r.t. the hidden dimension) provides a high-probability bound. In contrast, our result constitutes an existence proof, demonstrating the existence of a multi-head attention layer that can exactly memorize data points matching our claimed lower bound. In other words, while their memorization guarantee may fail with a certain probability, our guarantee stands firm with our construction.
> >
> >
> > [1] E. B. Baum. On the capabilities of multilayer perceptrons. 1988.
> >
> > [2] R. Vershynin. Memory capacity of neural networks with threshold and rectified linear unit activations. SIAM J., 2020.
> >
> > [3] G. Vardi, et al. On the optimal memorization power of reLU neural networks. ICLR, 2022.
> >
> > [4] C. Yun, et al. Small relu networks are powerful memorizers: a tight analysis of memorization capacity. NeurIPS, 2019.
> >
> > [5] N. Carlini, et al. Extracting Training Data from Large Language Models. USENIX, 2021.
> >
> > [6] S. Bubeck, et al. Network size and size of the weights in memorization with two-layers neural networks.

---

### Author Response · Authors · 2023-11-16
**General Response to Reviewers and Area Chair**

We thank all the reviewers for their constructive feedback. We are encouraged that the reviewers recognized the importance of our findings (R1), the contributions to a deeper understanding of transformers (R2), the introduction and verification of our set of more relaxed assumptions (R1, R2, R3, R4), and that our paper is well-written, well-organized and easy to follow (R1, R2, R3).

To address two questions raised by reviewers about the relationship between the "memorization" definition used in our paper compared to "associative memory" and "neural memory", we have updated our manuscript with a clarification footnote that the definition of memorization we use is distinct from the aforementioned concepts (please also see individual responses).

---

### Meta-Review · Area_Chair_rLoB · 2023-12-12

**Metareview:**

This paper presents a new lower bound on the memorization capacity of multi-head attention layers under a set of input-data assumptions. The results demonstrate that memorization increases with the number of heads, monotonically with the context size, and monotonically with the head dimension up to the context size. These findings are supported by experimental results obtained using synthetic data.

The analysis and results presented in this paper contribute significantly to our understanding of transformer architectures. However, it is important to note that the study primarily focuses on single-layer MHA, and expanding to multi-layered architectures could provide more comprehensive insights into the workings of these systems.

**Justification For Why Not Higher Score:**

N/A

**Justification For Why Not Lower Score:**

This paper presents a new lower bound on the memorization capacity of multi-head attention layers under a set of input-data assumptions. The results demonstrate that memorization increases with the number of heads, monotonically with the context size, and monotonically with the head dimension up to the context size. These findings are supported by experimental results obtained using synthetic data. The analysis and results presented in this paper contribute significantly to our understanding of transformer architectures.

---

### Decision · Program_Chairs · 2024-01-16

Accept (spotlight)